# SPARSE GAUSSIAN PROCESS VARIATIONAL AUTOEN-CODERS

## ABSTRACT

Large, multi-dimensional spatio-temporal datasets are omnipresent in modern science and engineering. An effective framework for handling such data are Gaussian process deep generative models (GP-DGMs), which employ GP priors over the latent variables of DGMs. Existing approaches for performing inference in GP-DGMs do not support sparse GP approximations based on inducing points, which are essential for the computational efficiency of GPs, nor do they handle missing data – a natural occurrence in many spatio-temporal datasets – in a principled manner. We address these shortcomings with the development of the sparse Gaussian process variational autoencoder (SGP-VAE), characterised by the use of partial inference networks for parameterising sparse GP approximations. Leveraging the benefits of amortised variational inference, the SGP-VAE enables inference in multi-output sparse GPs on previously unobserved data with no additional training. The SGP-VAE is evaluated in a variety of experiments where it outperforms alternative approaches including multi-output GPs and structured VAEs.

## 1 INTRODUCTION

Increasing amounts of large, multi-dimensional datasets that exhibit strong spatio-temporal dependencies are arising from a wealth of domains, including earth, social and environmental sciences (Atluri et al., 2018). For example, consider modelling daily atmospheric measurements taken by weather stations situated across the globe. Such data are (1) large in number; (2) subject to strong spatio-temporal dependencies; (3) multi-dimensional; and (4) non-Gaussian with complex dependencies across outputs. There exist two venerable approaches for handling these characteristics: Gaussian process (GP) regression and deep generative models (DGMs). GPs provide a framework for encoding high-level assumptions about latent processes, such as smoothness or periodicity, making them effective in handling spatio-temporal dependencies. Yet, existing approaches do not support the use of flexible likelihoods necessary for modelling complex multi-dimensional outputs. In contrast, DGMs support the use of flexible likelihoods; however, they do not provide a natural route through which spatio-temporal dependencies can be encoded. The amalgamation of GPs and DGMs, GP-DGMs, use latent functions drawn independently from GPs, which are then passed through a DGM at each input location. GP-DGMs combine the complementary strengths of both approaches, making them naturally suited for modelling spatio-temporal datasets.

Intrinsic to the application of many spatio-temporal datasets is the notion of *tasks*. For instance: medicine has individual patients; each trial in a scientific experiment produces an individual dataset; and, in the case of a single large dataset, it is often convenient to split it into separate tasks to improve computational efficiency. GP-DGMs support the presence of multiple tasks in a memory efficient way through the use of amortisation, giving rise to the Gaussian process variational autoencoder (GP-VAE), a model that has recently gained considerable attention from the research community (Pearce, 2020; Fortuin et al., 2020; Casale et al., 2018; Campbell & Liò, 2020; Ramchandran et al., 2020). However, previous work does not support sparse GP approximations based on inducing points, a necessity for modelling even moderately sized datasets. Furthermore, many spatio-temporal datasets contain an abundance of missing data: weather measurements are often absent due to sensor failure, and in medicine only single measurements are taken at any instance. Handling partial observations in a principled manner is essential for modelling spatio-temporal data, but is yet to be considered.

Our key technical contributions are as follows:

i) We develop the sparse GP-VAE (SGP-VAE), which uses inference networks to parameterise multi-output sparse GP approximations.

ii) We employ a suite of partial inference networks for handling missing data in the SGP-VAE.

iii) We conduct a rigorous evaluation of the SGP-VAE in a variety of experiments, demonstrating excellent performance relative to existing multi-output GPs and structured VAEs.

## 2  A FAMILY OF SPATIO-TEMPORAL VARIATIONAL AUTOENCODERS

Consider the multi-task regression problem in which we wish to model $T$ datasets $\mathcal{D} = \{\mathcal{D}^{(t)}\}_{t=1}^T$, each of which comprises input/output pairs $\mathcal{D}^{(t)} = \{\mathbf{x}_n^{(t)}, \mathbf{y}_n^{(t)}\}_{n=1}^{N_t}$, $\mathbf{x}_n^{(t)} \in \mathbb{R}^D$ and $\mathbf{y}_n^{(t)} \in \mathbb{R}^P$. Further, let any possible permutation of observed values be potentially missing, such that each observation $\mathbf{y}_n^{(t)} = \mathbf{y}_n^{o\,(t)} \cup \mathbf{y}_n^{u\,(t)}$ contains a set of observed values $\mathbf{y}_n^{o\,(t)}$ and unobserved values $\mathbf{y}_n^{u\,(t)}$, with $\mathcal{O}_n^{(t)}$ denoting the index set of observed values. For each task, we model the distribution of each observation $\mathbf{y}_n^{(t)}$, conditioned on a corresponding latent variable $\mathbf{f}_n^{(t)} \in \mathbb{R}^K$, as a fully-factorised Gaussian distribution parameterised by passing $\mathbf{f}_n^{(t)}$ through a decoder deep neural network (DNN) with parameters $\theta_2$. The elements of $\mathbf{f}_n^{(t)}$ correspond to the evaluation of a $K$-dimensional latent function $f^{(t)} = (f_1^{(t)}, f_2^{(t)}, \ldots, f_K^{(t)})$ at input $\mathbf{x}_n^{(t)}$. That is, $\mathbf{f}_n^{(t)} = f^{(t)}(\mathbf{x}_n^{(t)})$. Each latent function $f^{(t)}$ is modelled as being drawn from $K$ independent GP priors with hyper-parameters $\theta_1 = \{\theta_{1,k}\}_{k=1}^K$, giving rise to the complete probabilistic model:

$$
f^{(t)} \sim \prod_{k=1}^K \underbrace{\mathcal{GP}\left(0, k_{\theta_{1,k}}\left(\mathbf{x}, \mathbf{x}'\right)\right)}_{p_{\theta_1}(f_k^{(t)})} \qquad \mathbf{y}^{(t)}|\mathbf{f}^{(t)} \sim \prod_{n=1}^{N_t} \underbrace{\mathcal{N}\left(\boldsymbol{\mu}_{\theta_2}^o(\mathbf{f}_n^{(t)}),\ \mathrm{diag}\,\boldsymbol{\sigma}_{\theta_2}^{o}{}^2(\mathbf{f}_n^{(t)})\right)}_{p_{\theta_2}(\mathbf{y}_n^{o\,(t)}|f^{(t)}, \mathbf{x}_n^{(t)}, \mathcal{O}_n^{(t)})} \tag{1}
$$

where $\boldsymbol{\mu}_{\theta_2}^o(\mathbf{f}_n^{(t)})$ and $\boldsymbol{\sigma}_{\theta_2}^{o}{}^2(\mathbf{f}_n^{(t)})$ are the outputs of the decoder indexed by $\mathcal{O}_n^{(t)}$. We shall refer to the set $\theta = \{\theta_1, \theta_2\}$ as the model parameters, which are shared across tasks. The probabilistic model in equation 1 explicitly accounts for dependencies between latent variables through the GP prior. The motive of the latent structure is twofold: to discover a simpler representation of each observation, and to capture the dependencies between observations at different input locations.

### 2.1  MOTIVATION FOR SPARSE APPROXIMATIONS AND AMORTISED INFERENCE

The use of amortised inference in DGMs and sparse approximations in GPs enables inference in these respective models to scale to large quantities of data. To ensure the same for the GP-DGM described in equation 1, we require the use of both techniques. In particular, amortised inference is necessary to prevent the number of variational parameters scaling with $\mathcal{O}\left(\sum_{t=1}^T N^{(t)}\right)$. Further, the inference network can be used to condition on previously unobserved data without needing to learn new variational parameters. Similarly, sparse approximations are necessary to prevent the computational complexity increasing cubically with the size of each task $\mathcal{O}\left(\sum_{t=1}^T N^{(t)^3}\right)$.

Unfortunately, it is far from straightforward to combine sparse approximations and amortised inference in a computationally efficient way. To see this, consider the standard form for the sparse GP approximate posterior, $q(f) = p_{\theta_1}(f_{\backslash \mathbf{u}}|\mathbf{u})q(\mathbf{u})$ where $q(\mathbf{u}) = \mathcal{N}(\mathbf{u}; \mathbf{m}, \mathbf{S})$ with $\mathbf{m}, \mathbf{S}$ and $\mathbf{Z}$, the inducing point locations, being the variational parameters. $q(\mathbf{u})$ does not decompose into a product over $N^{(t)}$ factors and is therefore not amendable to per-datapoint amortisation. That is, $\mathbf{m}$ and $\mathbf{S}$ must be optimised as free-form variational parameters. A naïve approach to achieving per-datapoint amortisation is to decompose $q(\mathbf{u})$ into the prior $p_{\theta_1}(\mathbf{u})$ multiplied by the product of approximate likelihoods, one for each inducing point. Each approximate likelihood is itself equal to the product of per-datapoint approximate likelihoods, which depend on both the observation $\mathbf{y}_n^o$ and the distance of the input $\mathbf{x}_n$ to that of the inducing point. An inference network which takes these two values of inputs can be used to obtain the parameters of the approximate likelihood factors. Whilst we found that this approach worked, it is somewhat unprincipled. Moreover, it requires passing each datapoint/inducing point pair through an inference network, which scales very poorly. In the following

section, we introduce a theoretically principled decomposition of $q(f)$ we term the *sparse structured approximate posterior* which will enable efficient amortization.

## 2.2 THE SPARSE STRUCTURED APPROXIMATE POSTERIOR

By simultaneously leveraging amortised inference and sparse GP approximations, we can perform efficient and scalable approximate inference. We specify the sparse structured approximate posterior, $q(f^{(t)})$, which approximates the intractable true posterior for task $t$:

$$
\begin{aligned}
p_\theta(f^{(t)}|\mathbf{y}^{(t)}, \mathbf{X}^{(t)}) &= \frac{1}{\mathcal{Z}_p} p_{\theta_1}(f^{(t)}) \prod_{n=1}^{N_t} p_{\theta_2}(\mathbf{y}_n^{o\,(t)}|f^{(t)}, \mathbf{x}_n^{(t)}, \mathcal{O}_n^{(t)}) \\
&\approx \frac{1}{\mathcal{Z}_q} p_{\theta_1}(f^{(t)}) \prod_{n=1}^{N_t} l_{\phi_l}(\mathbf{u}; \mathbf{y}_n^{o\,(t)}, \mathbf{x}_n^{(t)}, \mathbf{Z}) = q(f^{(t)}).
\end{aligned}
\tag{2}
$$

Analogous to its presence in the true posterior, the approximate posterior retains the GP prior, yet replaces each non-conjugate likelihood factor with an *approximate likelihood*, $l_{\phi_l}(\mathbf{u}; \mathbf{y}_n^{o\,(t)}, \mathbf{x}_n^{(t)}, \mathbf{Z})$, over a set of $KM$ 'inducing points', $\mathbf{u} = \cup_{k=1}^K \cup_{m=1}^M u_{mk}$, at 'inducing locations', $\mathbf{Z} = \cup_{k=1}^K \cup_{m=1}^M \mathbf{z}_{mk}$. For tractability, we restrict the approximate likelihoods to be Gaussians factorised across each latent dimension, parameterised by passing each observation through a *partial inference network*:

$$
l_{\phi_l}(\mathbf{u}_k; \mathbf{y}_n^{o\,(t)}, \mathbf{x}_n^{(t)}, \mathbf{Z}_k) = \mathcal{N}\left(\mu_{\phi_l,k}(\mathbf{y}_n^{o\,(t)}); \; k_{f_{nk}^{(t)}\mathbf{u}_k} \mathbf{K}_{\mathbf{u}_k\mathbf{u}_k}^{-1} \mathbf{u}_k, \; \sigma_{\phi_l,k}^2(\mathbf{y}_n^{o\,(t)})\right)
\tag{3}
$$

where $\phi_l$ denotes the weights and biases of the partial inference network, whose outputs are shown in red. This form is motivated by the work of Bui et al. (2017), who demonstrate the optimality of approximate likelihoods of the form $\mathcal{N}\left(g_n; \; k_{f_{nk}^{(t)}\mathbf{u}_k} \mathbf{K}_{\mathbf{u}_k\mathbf{u}_k}^{-1} \mathbf{u}_k, \; v_n\right)$, a result we prove in Appendix A.1. Whilst, in general, the optimal free-form values of $g_n$ and $v_n$ depend on all of the data points, we make the simplifying assumption that they depend only on $\mathbf{y}_n^{o\,(t)}$. For GP regression with Gaussian noise, this assumption holds true as $g_n = y_n$ and $v_n = \sigma_y^2$ (Bui et al., 2017).

The resulting approximate posterior can be interpreted as the exact posterior induced by a surrogate regression problem, in which 'pseudo-observations' $g_n$ are produced from a linear transformation of inducing points with additive 'pseudo-noise' $v_n$, $g_n = k_{f_{nk}^{(t)}\mathbf{u}_k} \mathbf{K}_{\mathbf{u}_k\mathbf{u}_k}^{-1} \mathbf{u}_k + \sqrt{v_n}\epsilon$. The inference network learns to construct this surrogate regression problem such that it results in a posterior that is close to our target posterior.

By sharing variational parameters $\phi = \{\phi_l, \mathbf{Z}\}$ across tasks, inference is amortised across both datapoints and tasks. The approximate posterior for a single task corresponds to the product of $K$ independent GPs, with mean and covariance functions

$$
\begin{aligned}
\hat{m}_k^{(t)}(\mathbf{x}) &= k_{f_k^{(t)}\mathbf{u}_k} \boldsymbol{\Phi}_k^{(t)} \mathbf{K}_{\mathbf{u}_k\mathbf{f}_k^{(t)}} {\boldsymbol{\Sigma}_{\phi_l,k}^{(t)}}^{-1} \boldsymbol{\mu}_{\phi_l,k}^{(t)} \\
\hat{k}_k^{(t)}(\mathbf{x}, \mathbf{x}') &= k_{f_k^{(t)}f_k'^{(t)}} - k_{f_k^{(t)}\mathbf{u}_k} \mathbf{K}_{\mathbf{u}_k\mathbf{u}_k}^{-1} k_{\mathbf{u}_k f_k'^{(t)}} + k_{f_k^{(t)}\mathbf{u}_k} \boldsymbol{\Phi}_k^{(t)} k_{\mathbf{u}_k f_k'^{(t)}}
\end{aligned}
\tag{4}
$$

where ${\boldsymbol{\Phi}_k^{(t)}}^{-1} = \mathbf{K}_{\mathbf{u}_k\mathbf{u}_k} + \mathbf{K}_{\mathbf{u}_k\mathbf{f}_k^{(t)}} {\boldsymbol{\Sigma}_{\phi_l,k}^{(t)}}^{-1} \mathbf{K}_{\mathbf{f}_k^{(t)}\mathbf{u}_k}$, $\left[\boldsymbol{\mu}_{\phi_l,k}^{(t)}\right]_i = \mu_{\phi_l,k}(\mathbf{y}_i^{o(t)})$ and $\left[\boldsymbol{\Sigma}_{\phi_l,k}^{(t)}\right]_{ij} = \delta_{ij}\sigma_{\phi_l,k}^2(\mathbf{y}_i^{o(t)})$. See Appendix A.2 for a complete derivation. The computational complexity associated with evaluating the mean and covariance functions is $\mathcal{O}\left(KM^2N^{(t)}\right)$, a significant improvement over the $\mathcal{O}\left(P^3{N^{(t)}}^3\right)$ cost associated with exact multi-output GPs for $KM^2 \ll P^3{N^{(t)}}^2$. We refer to the combination of the aforementioned probabilistic model and sparse structured approximate posterior as the SGP-VAE.

The SGP-VAE addresses three major shortcomings of existing sparse GP frameworks. First, the inference network can be used to condition on previously unobserved data without needing to learn new variational parameters. Suppose we use the standard sparse GP variational approximation $q(f) = p_{\theta_1}(f_{\backslash\mathbf{u}}|\mathbf{u})q(\mathbf{u})$ where $q(\mathbf{u}) = \mathcal{N}(\mathbf{u}; \mathbf{m}, \mathbf{S})$. If more data are observed, $\mathbf{m}$ and $\mathbf{S}$ have to be re-optimised. When an inference network is used to parameterise $q(\mathbf{u})$, the approximate posterior

is 'automatically' updated by mapping from the new observations to their corresponding approximate likelihood terms. Second, the complexity of the approximate posterior can be modified as desired with no changes to the inference network, or additional training, necessary: any changes in the morphology of inducing points corresponds to a deterministic transformation of the inference network outputs. Third, if the inducing point locations are fixed, then the number of variational parameters does not depend on the size of the dataset, even as more inducing points are added. This contrasts with the standard approach, in which new variational parameters are appended to $\mathbf{m}$ and $\mathbf{S}$.

## 2.3 TRAINING THE SGP-VAE

Learning and inference in the SGP-VAE are concerned with determining the model parameters $\theta$ and variational parameters $\phi$. These objectives can be attained simultaneously by maximising the *evidence lower bound* (ELBO), given by $\mathcal{L}_{\text{ELBO}} = \sum_{t=1}^{T} \mathcal{L}_{\text{ELBO}}^{(t)}$ where

$$\mathcal{L}_{\text{ELBO}}^{(t)} = \mathbb{E}_{q(f^{(t)})} \left[ \frac{p_\theta(\mathbf{y}^{(t)}, f^{(t)})}{q(f^{(t)})} \right] = \mathbb{E}_{q(\mathbf{f}^{(t)})} \left[ \log p_\theta(\mathbf{y}^{(t)}|\mathbf{f}^{(t)}) \right] - \text{KL} \left( q^{(t)}(\mathbf{u}) \, \| \, p_{\theta_1}(\mathbf{u}) \right) \quad (5)$$

and $q^{(t)}(\mathbf{u}) \propto p_{\theta_1}(\mathbf{u}) \prod_{n=1}^{N_t} l_{\phi_l}(\mathbf{u}; \mathbf{y}_n^{o\,(t)}, \mathbf{x}_n^{(t)}, \mathbf{Z})$. Fortunately, since both $q^{(t)}(\mathbf{u})$ and $p_{\theta_1}(\mathbf{u})$ are multivariate Gaussians, the final term, and its gradients, has an analytic solution. The first term amounts to propagating a Gaussian through a non-linear DNN, so must be approximated using a Monte Carlo estimate. We employ the reparameterisation trick (Kingma & Welling, 2014) to account for the dependency of the sampling procedure on both $\theta$ and $\phi$ when estimating its gradients.

We mini-batch over tasks, such that only a single $\mathcal{L}_{\text{ELBO}}^{(t)}$ is computed per update. Importantly, in combination with the inference network, this means that we avoid having to retain the $\mathcal{O}\left(TM^2\right)$ terms associated with $T$ Cholesky factors if we were to use a free-form $q(\mathbf{u})$ for each task. Instead, the memory requirement is dominated by the $\mathcal{O}\left(KM^2 + KNM + |\phi_l|\right)$ terms associated with storing $\mathbf{K}_{\mathbf{u}_k \mathbf{u}_k}$, $\mathbf{K}_{\mathbf{u}_k \mathbf{f}_k^{(t)}}$ and $\phi_l$, as instantiating $\boldsymbol{\mu}_{\phi_l,k}^{(t)}$ and $\boldsymbol{\Sigma}_{\phi_l,k}^{(t)}$ involves only $\mathcal{O}\left(KN\right)$ terms.[1] This corresponds to a considerable reduction in memory. See Appendix C for a thorough comparison of memory requirements.

## 2.4 PARTIAL INFERENCE NETWORKS

Partially observed data is regularly encountered in spatio-temporal datasets, making it necessary to handle it in a principled manner. Missing data is naturally handled by Bayesian inference. However, for models using inference networks, it necessitates special treatment. One approach to handling partially observed data is to impute missing values with zeros (Nazabal et al., 2020; Fortuin et al., 2020). Whilst simple to implement, zero imputation is theoretically unappealing as the inference network can no longer distinguish between a missing value and a true zero.

Instead, we turn towards the ideas of Deep Sets (Zaheer et al., 2017). By coupling the observed value with dimension index, we may reinterpret each partial observation as a permutation invariant set. We define a family of permutation invariant partial inference networks[2] as

$$\left( \boldsymbol{\mu}_\phi(\mathbf{y}_n^o), \, \log \boldsymbol{\sigma}_\phi^2(\mathbf{y}_n^o) \right) = \rho_{\phi_2} \left( \sum_{p \in \mathcal{O}_n} h_{\phi_1}(\mathbf{s}_{np}) \right) \quad (6)$$

where $h_{\phi_1} : \mathbb{R}^2 \to \mathbb{R}^R$ and $\rho_{\phi_2} : \mathbb{R}^R \to \mathbb{R}^{2P}$ are DNN mappings with parameters $\phi_1$ and $\phi_2$, respectively. $\mathbf{s}_{np}$ denotes the couples of observed value $y_{np}$ and corresponding dimension index $p$. The formulation in equation 6 is identical to the partial variational autoencoder (VAE) framework established by Ma et al. (2019). There are a number of partial inference networks which conform to this general framework, three of which include:

---

[1]This assumes input locations are shared across tasks, which is true for all experiments we considered.

[2]Whilst the formulation in equation 6 can describe *any* permutation invariant set function, there is a caveat: both $h_{\phi_1}$ and $\rho_{\phi_2}$ can be infinitely complex, hence linear complexity is not guaranteed.

**PointNet** Inspired by the PointNet approach of Qi et al. (2017) and later developed by Ma et al. (2019) for use in partial VAEs, the PointNet specification uses the concatenation of dimension index with observed value: $\mathbf{s}_{np} = (p, y_{np})$. This specification treats the dimension indices as continuous variables. Thus, an implicit assumption of PointNet is the assumption of smoothness between values of neighbouring dimensions. Although valid in a computer vision application, it is ill-suited for tasks in which the indexing of dimensions is arbitrary.

**IndexNet** Alternatively, one may use the dimension index to select the first DNN mapping: $h_{\phi_1}(\mathbf{s}_{np}) = h_{\phi_{1,p}}(y_{np})$. Whereas PointNet treats dimension indices as points in space, this specification retains their role as indices. We refer to it as the IndexNet specification.

**FactorNet** A special case of IndexNet, first proposed by Vedantam et al. (2017), uses a separate inference network for each observation dimension. The approximate likelihood is factorised into a product of Gaussians, one for each output dimension: $l_{\phi_l}(\mathbf{u}_k; \mathbf{y}_n^o, \mathbf{x}_n, \mathbf{Z}_k) = \prod_{p \in \mathcal{O}_n} \mathcal{N}\left(\mu_{\phi_l, p_k}(y_{np}); k_{f_{nk}\mathbf{u}_k}\mathbf{K}_{\mathbf{u}_k\mathbf{u}_k}^{-1}\mathbf{u}_k, \sigma_{\phi_l, p_k}^2(y_{np})\right)$. We term this approach FactorNet.

See Appendix G for corresponding computational graphs. Note that FactorNet is equivalent to IndexNet with $\rho_{\phi_2}$ defined by the deterministic transformations of natural parameters of Gaussian distributions. Since IndexNet allows this transformation to be learnt, we might expect it to always produce a better partial inference network for the task at hand. However, it is important to consider the ability of inference networks to generalise. Although more complex inference networks will better approximate the optimal non-amortised approximate posterior on training data, they may produce poor approximations to it on the held-out data.[3] In particular, FactorNet is constrained to consider the individual contribution of each observation dimension, whereas the others are not. Doing so is necessary for generalising to different quantities and patterns of missingness, hence we anticipate FactorNet to perform better in such settings.

## 3 Related Work

We focus our comparison on approximate inference techniques; however, Appendix D presents a unifying view of GP-DGMs.

**Structured Variational Autoencoder** Only recently has the use of structured latent variable priors in VAEs been considered. In their seminal work, Johnson et al. (2016) investigate the combination of probabilistic graphical models with neural networks to learn structured latent variable representations. The authors consider a two stage iterative procedure, whereby the optimum of a surrogate objective function — containing approximate likelihoods in place of true likelihoods — is found and substituted into the original ELBO. The resultant structured VAE (SVAE) objective is then optimised. In the case of fixed model parameters $\theta$, the SVAE objective is equivalent to optimising the ELBO using the structured approximate posterior over latent variables $q(\mathbf{z}) \propto p_\theta(\mathbf{z})l_\phi(\mathbf{z}|\mathbf{y})$. Accordingly, the SGP-VAE can be viewed as an instance of the SVAE. Lin et al. (2018) build upon the SVAE, proposing a structured approximate posterior of the form $q(\mathbf{z}) \propto q_\phi(\mathbf{z})l_\phi(\mathbf{z}|\mathbf{y})$. The authors refer to the approximate posterior as the structured inference network (SIN). Rather than using the latent prior $p_\theta(\mathbf{z})$, SIN incorporates the model's latent structure through $q_\phi(\mathbf{z})$. The core advantage of SIN is its extension to more complex latent priors containing non-conjugate factors — $q_\phi(\mathbf{z})$ can replace them with their nearest conjugate approximations whilst retaining a similar latent structure. Although the frameworks proposed by Johnson et al. and Lin et al. are more general than ours, the authors only consider Gaussian mixture model and linear dynamical system (LDS) latent priors.

**Gaussian Process Variational Autoencoders** The earliest example of combining VAEs with GPs is the GP prior VAE (GPPVAE) (Casale et al., 2018). There are significant differences between our work and the GPPVAE, most notably in the GPPVAE's use of a fully-factorised approximate posterior — an approximation that is known to perform poorly in time-series and spatial settings (Turner & Sahani, 2011). Closely related to the GPPVAE is Ramchandran et al.'s (2020) longitudinal VAE, which also adopts a fully-factorised approximate posterior, yet uses additive covariance functions for heterogeneous input data. Fortuin et al. (2020) consider the use of a Gaussian ap-

---

[3]This kind of 'overfitting' is different to the classical notion of overfitting model parameters. It refers to how well optimal non-amortised approximate posteriors are approximated on the training versus test data.

proximate posterior with a tridiagonal precision matrix $\mathbf{\Lambda}$, $q(\mathbf{f}) = \mathcal{N}\left(\mathbf{f}; \mathbf{m}, \mathbf{\Lambda}^{-1}\right)$, where $\mathbf{m}$ and $\mathbf{\Lambda}$ are parameterised by an inference network. Whilst this permits computational efficiency, the parameterisation is only appropriate for regularly spaced temporal data and neglects rigorous treatment of long term dependencies. Campbell & Liò (2020) employ an equivalent sparsely structured variational posterior as that used by Fortuin et al., extending the framework to handle more general spatio-temporal data. Their method is similarly restricted to regularly spaced spatio-temporal data. A fundamental difference between our framework and that of Fortuin et al. and Campbell & Liò is the inclusion of the GP prior in the approximate posterior. As shown by Opper & Archambeau (2009), the structured approximate posterior is identical in form to the optimum Gaussian approximation to the true posterior. Most similar to ours is the approach of Pearce (2020), who considers the structured approximate posterior $q(f) = \frac{1}{\mathcal{Z}_q} p_{\theta_1}(f) \prod_{n=1}^{N} l_{\phi_l}(\mathbf{f}_n; \mathbf{y}_n)$. We refer to this as the GP-VAE. Pearce's approach is a special case of the SGP-VAE for $\mathbf{u} = \mathbf{f}$ and no missing data. Moreover, Pearce only considers the application to modelling pixel dynamics and the comparison to the standard VAE. See Appendix B for further details.

## 4 EXPERIMENTS

We investigate the performance of the SGP-VAE in illustrative bouncing ball experiments, followed by experiments in the small and large data regimes. The first bouncing ball experiment provides a visualisation of the mechanics of the SGP-VAE, and a quantitative comparison to other structured VAEs. The proceeding small-scale experiments demonstrate the utility of the GP-VAE and show that amortisation, especially in the presence of partially observed data, is not at the expense of predictive performance. In the final two experiments, we showcase the efficacy of the SGP-VAE on large, multi-output spatio-temporal datasets for which the use of amortisation is necessary. Full experimental details are provided in Appendix E.

### 4.1 SYNTHETIC BOUNCING BALL EXPERIMENT

The bouncing ball experiment — first introduced by Johnson et al. (2016) for evaluating the SVAE and later considered by Lin et al. (2018) for evaluating SIN — considers a sequence of one-dimensional images of height 10 representing a ball bouncing under linear dynamics, $(\mathbf{x}_n^{(t)} \in \mathbb{R}^1, \mathbf{y}_n^{(t)} \in \mathbb{R}^{10})$. The GP-VAE is able to significantly outperform both the SVAE and SIN in the original experiment, as shown in Figure 1a. To showcase the versatility of the SGP-VAE, we extend the complexity of the original experiment to consider a sequence of images of height 100, $\mathbf{y}_n^{(t)} \in \mathbb{R}^{100}$, representing two bouncing balls: one under linear dynamics and another under gravity. Furthermore, the images are corrupted by removing 25% of the pixels at random. The dataset consists of $T = 80$ noisy image sequences, each of length $N = 500$, with the goal being to predict the trajectory of the ball given a prefix of a longer sequence.

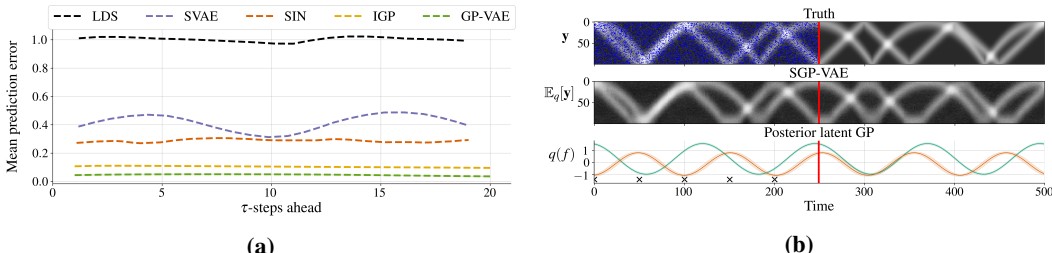

**(a)**  **(b)**

**Figure 1: a)** Comparing the GP-VAE's predictive performance to that of the SVAE, SIN, an LDS and independent GPs (IGP). **b)** Top: sequence of images representing two bouncing balls. Middle: mean of the SGP-VAE's predictive distribution conditioned on partial observations up to the red line. Bottom: latent approximate GPs posterior, alongside the inducing point locations (crosses).

Using a two-dimensional latent space with periodic kernels, Figure 1b compares the posterior latent GPs and the mean predictive distribution with the ground truth for a single image sequence. Observe that the SGP-VAE has 'disentangled' the dynamics of each ball, using a single latent dimension to

model each. The SGP-VAE reproduces the image sequences with impressive precision, owing in equal measure to (1) the ability of the GPs prior to model the latent dynamics and (2) the flexibility of the likelihood function to map to the high-dimensional observations.

## 4.2 SMALL-SCALE EXPERIMENTS

**EEG** Adopting the experimental procedure laid out by Requeima et al. (2019), we consider an EEG dataset consisting of $N = 256$ measurements taken over a one second period. Each measurement comprises voltage readings taken by seven electrodes, FZ and F1-F6, positioned on the patient's scalp ($\mathbf{x}_n \in \mathbb{R}^1$, $\mathbf{y}_n \in \mathbb{R}^7$). The goal is to predict the final 100 samples for electrodes FZ, F1 and F2 having observed the first 156 samples, as well as all 256 samples for electrodes F3-F6.

**Jura** The Jura dataset is a geospatial dataset comprised of $N = 359$ measurements of the topsoil concentrations of three heavy metals — Cadmium Nickel and Zinc — collected from a 14.5km$^2$ region of the Swiss Jura ($\mathbf{x}_n \in \mathbb{R}^2$, $\mathbf{y}_n \in \mathbb{R}^3$) (Goovaerts, 1997). Adopting the experimental procedure laid out by others (Goovaerts, 1997; Álvarez & Lawrence, 2011; Requeima et al., 2019), the dataset is divided into a training set consisting of Nickel and Zinc measurements for all 359 locations and Cadmium measurements for just 259 locations. Conditioned on the observed training set, the goal is to predict the Cadmium measurements at the remaining 100 locations.

**Table 1:** A comparison between multi-output GP models on the EEG and Jura experiments.

| | **Metric** | IGP[†] | GPAR[†] | GP-VAE | | | | |
| | | | | ZI | PointNet | IndexNet | FactorNet | FF |
| --- | --- | --- | --- | --- | --- | --- | --- | --- |
| EEG | SMSE | 1.75 | **0.26** | **0.27 (0.03)** | 0.60 (0.09) | **0.24 (0.02)** | **0.28 (0.04)** | **0.25 (0.02)** |
| | NLL | 2.60 | **1.63** | 2.24 (0.37) | 3.03 (1.34) | 2.01 (0.28) | 2.23 (0.21) | 2.13 (0.28) |
| Jura | MAE | 0.57 | **0.41** | **0.42 (0.01)** | 0.45 (0.02) | 0.44 (0.02) | **0.40 (0.01)** | **0.41 (0.01)** |
| | NLL | - | - | **1.13 (0.09)** | **1.13 (0.08)** | **1.12 (0.08)** | **1.00 (0.06)** | **1.04 (0.06)** |

[†]Results taken directly from Requeima et al. (2019).

Table 1 compares the performance of the GP-VAE using the three partial inference networks presented in Section 2.4, as well as zero imputation (ZI), with independent GPs (IGP) and the GP autoregressive regression model (GPAR), which, to our knowledge, has the strongest published performance on these datasets. We also give the results for the best performing GP-VAE[4] using a non-amortised, or 'free-form' (FF), approximate posterior, with model parameters $\theta$ kept fixed to the optimum found by the amortised GP-VAE and variational parameters initialised to the output of the optimised inference network. All GP-VAE models use a two- and three-dimensional latent space for EEG and Jura, respectively, with squared exponential (SE) kernels. The results highlight the poor performance of independent GPs relative to multi-output GPs, demonstrating the importance of modelling output dependencies. The GP-VAE achieves impressive SMSE and MAE[5] on the EEG and Jura datasets using all partial inference networks except for PointNet. In Appendix F we demonstrate superior performance of the GP-VAE relative to the GPPVAE, which can be attributed to the use of the structured approximate posterior over the mean-field approximate posterior used by the GPPVAE. Importantly, the negligible difference between the results using free-form and amortised approximate posteriors indicates that amortisation is not at the expense of predictive performance.

Whilst GPAR performs as strongly as the GP-VAE in the small-scale experiments above, it has two key limitations which severely limit the types of applications where it can be used. First, it can only be used with specific patterns of missing data and not when the pattern of missingness is arbitrary. Second, it is not scalable and would require further development to handle the large datasets considered in this paper. In contrast, the SGP-VAE is far more flexible: it handles arbitrary patterns of missingness, and scales to large number of datapoints and tasks. A distinct advantage of the SGP-VAE is that it models $P$ outputs with just $K$ latent GPs. This differs from GPAR, which uses $P$

---

[4]i.e. using IndexNet for EEG and FactorNet for Jura.

[5]The two different performance metrics are adopted to enable a comparison to the results of Requeima et al..

GPs. Whilst this is not an issue for the small-scale experiments, it quickly becomes computationally burdensome when $P$ becomes large. The true efficacy of the SGP-VAE is demonstrated in the following three experiments, where the number of datapoints and tasks is large, and the patterns of missingness are random.

### 4.3 LARGE-SCALE EEG EXPERIMENT

We consider an alternative setting to the original small-scale EEG experiment, in which the datasets are formed from $T = 60$ recordings of length $N = 256$, each with 64 observed voltage readings ($\mathbf{y}_n \in \mathbb{R}^{64}$). For each recording, we simulated electrode 'blackouts' by removing consecutive samples at random. We consider two experiments: in the first, we remove equal 50% of data from both the training and test datasets; in the second, we remove 10% of data from the training dataset and 50% from the test dataset. Both experiments require the partial inference network to generalise to different patterns of missingness, with the latter also requiring generalisation to different quantities of missingness. Each model is trained on 30 recordings, with the predictive performance assessed on the remaining 30 recordings. Figure 2 compares the performance of the SGP-VAE with that of independent GPs as the number of inducing points varies, with $M = 256$ representing use of the GP-VAE. In each case, we use a 10-dimensional latent space with SE kernels. The SGP-VAE using PointNet results in substantially worse performance than the other partial inference networks, achieving an average SMSE and NLL of 1.30 and 4.05 on the first experiment for $M = 256$. Similarly, using a standard VAE results in poor performance, achieving an average SMSE and NLL of 1.62 and 3.48. These results are excluded from Figure 2 for the sake of readability.

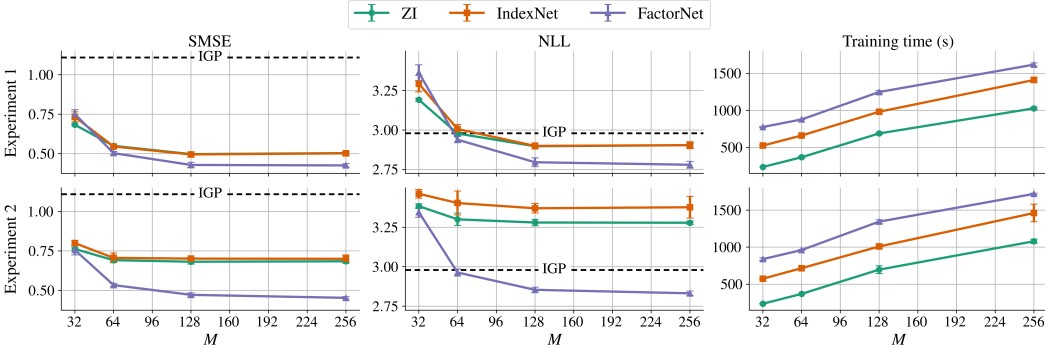

**Figure 2:** Variation in performance of the SGP-VAE on the large-scale EEG experiment as the number of inducing points, $M$, varies.

For all partial inference networks, the SGP-VAE achieves a significantly better SMSE than independent GPs in both experiments, owing to its ability to model both input and output dependencies. For the first experiment, the performance using FactorNet is noticeably better than using either IndexNet or zero imputation; however, this comes at the cost of a greater computational complexity associated with learning an inference network for each output dimension. Whereas the performance for the SGP-VAE using IndexNet and zero imputation significantly worsens on the second experiment, the performance using FactorNet is comparable to the first experiment. This suggests it is the only partial inference network that is able to accurately quantify the contribution of each output dimension to the latent posterior, enabling it to generalise to different quantities of missing data.

The advantages of using a sparse approximation are clear — using $M = 128$ inducing points results in a slightly worse average SMSE and NLL, yet significantly less computational cost.

### 4.4 JAPANESE WEATHER EXPERIMENT

Finally, we consider a dataset comprised of 731 daily climate reports from 156 Japanese weather stations throughout 1980 and 1981, a total of 114,036 multi-dimensional observations. Weather reports consist of a date and location, including elevation, alongside the day's maximum, minimum and average temperature, precipitation and snow depth ($\mathbf{x}_n^{(t)} \in \mathbb{R}^4$, $\mathbf{y}_n^{(t)} \in \mathbb{R}^5$), any number of which is potentially missing. We treat each week as a single task, resulting in $T = 105$ tasks with

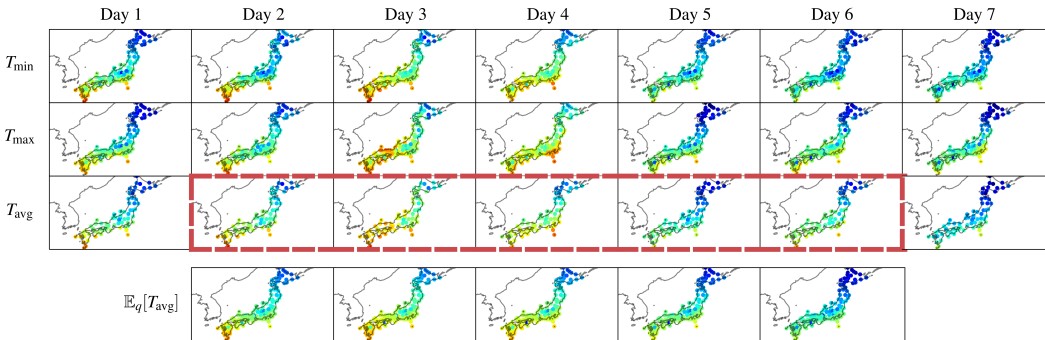

**Figure 3:** An illustration of the Japanese weather experiment. The dotted red lines highlight the missing data, with the SGP-VAE's predictive mean shown below.

$N = 1092$ data points each. The goal is to predict the average temperature for all stations on the middle five days, as illustrated in Figure 3. Each model is trained on all the data available from 1980. For evaluation, we use data from both 1980 and 1981 with additional artificial missingness — the average temperature for the middle five days and a random 25% of minimum and maximum temperature measurements[6]. Similar to the second large-scale EEG experiment, the test datasets have more missing data than the training datasets. Table 2 compares the performance of the SGP-VAE using 100 inducing points to that of a standard VAE using FactorNet and a baseline of mean imputation. All models use a three-dimensional latent space with SE kernels.

**Table 2:** A comparison between model performance on the Japanese weather experiment.

| | Metric | MI | IGP | VAE | SGP-VAE | | | |
| | | | | | ZI | PointNet | IndexNet | FactorNet |
|---|---|---|---|---|---|---|---|---|
| 1980 | RMSE | 9.16 | 2.49 (0.00) | 2.31 (0.35) | 3.21 (0.35) | 3.53 (0.13) | 2.59 (0.11) | **1.60 (0.08)** |
| | NLL | - | 2.84 (0.00) | 2.68 (0.15) | 2.68 (0.15) | 2.80 (0.05) | 2.44 (0.04) | **2.18 (0.02)** |
| 1981 | RMSE | 9.27 | 2.41 (0.00) | 3.20 (0.37) | 3.20 (0.37) | 3.61 (0.14) | 2.55 (0.11) | **1.68 (0.09)** |
| | NLL | - | 2.73 (0.00) | 2.67 (0.15) | 2.67 (0.15) | 2.83 (0.06) | 2.43 (0.04) | **2.20 (0.02)** |

All models significantly outperform the mean imputation baseline (MI) and are able to generalise inference to the unseen 1981 dataset without any loss in predictive performance. The SGP-VAE achieves better predictive performance than both the standard VAE and independent GPs, showcasing its effectiveness in modelling large spatio-temporal datasets. The SGP-VAE using FactorNet achieves the best predictive performance on both datasets. The results indicate that FactorNet is the only partial inference network capable of generalising to different quantities and patterns of missingness, supporting the hypothesis made in Section 2.4.

## 5 CONCLUSION

The SGP-VAE is a scalable approach to training GP-DGMs which combines sparse inducing point methods for GPs and amortisation for DGMs. The approach is ideally suited to spatio-temporal data with missing observations, where it outperforms VAEs and multi-output GPs. Future research directions include generalising the framework to leverage state-space GP formulations for additional scalability and applications to streaming multi-output data.

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

# A  MATHEMATICAL DERIVATIONS

## A.1  OPTIMALITY OF APPROXIMATE LIKELIHOODS

To simplify notation, we shall consider the case $P = 1$ and $K = 1$. Separately, Opper & Archambeau (2009) considered the problem of performing variational inference in a GP for non-Gaussian likelihoods. They consider a multivariate Gaussian approximate posterior, demonstrating that the optimal approximate posterior takes the form

$$q(\mathbf{f}) = \frac{1}{\mathcal{Z}} p(\mathbf{f}) \prod_{n=1}^{N} \mathcal{N}\left(f_n;\ g_n,\ v_n\right), \tag{7}$$

requiring a total of $2N$ variational parameters ($\{g_n, v_n\}_{n=1}^{N}$).

In this section, we derive a result that generalises this to inducing point approximations, showing that for fixed $M$ the optimal approximate posterior can be represented by $\max(M(M+1)/2 + M,\ 2N)$. Following Titsias (2009), we consider an approximate posterior of the form

$$q(f) = q(\mathbf{u})p(f_{\setminus \mathbf{u}}|\mathbf{u}) \tag{8}$$

where $q(\mathbf{u}) = \mathcal{N}\left(\mathbf{u};\ \hat{\mathbf{m}}_{\mathbf{u}},\ \hat{\mathbf{K}}_{\mathbf{uu}}\right)$ is constrained to be a multivariate Gaussian with mean $\hat{\mathbf{m}}_{\mathbf{u}}$ and covariance $\hat{\mathbf{K}}_{\mathbf{uu}}$. The ELBO is given by

$$
\begin{aligned}
\mathcal{L}_{\text{ELBO}} &= \mathbb{E}_{q(\mathbf{f})}\left[\log p(\mathbf{y}|\mathbf{f})\right] - \text{KL}\left(q(\mathbf{u}) \,\|\, p(\mathbf{u})\right) \\
&= \mathbb{E}_{q(\mathbf{u})}\left[\mathbb{E}_{p(\mathbf{f}|\mathbf{u})}\left[\log p(\mathbf{y}|\mathbf{f})\right]\right] - \text{KL}\left(q(\mathbf{u}) \,\|\, p(\mathbf{u})\right) \\
&= \sum_{n=1}^{N} \mathbb{E}_{q(\mathbf{u})}\left[\mathbb{E}_{\mathcal{N}\left(f_n;\ \mathbf{A}_n\mathbf{u}+a_n,\ \mathbf{K}_{f_n|\mathbf{u}}\right)}\left[\log p(\mathbf{y}_n|f_n)\right]\right] - \text{KL}\left(q(\mathbf{u}) \,\|\, p(\mathbf{u})\right)
\end{aligned}
\tag{9}
$$

where

$$\mathbf{A}_n = \mathbf{K}_{f_n\mathbf{u}}\mathbf{K}_{\mathbf{uu}}^{-1} \tag{10}$$

$$a_n = m_{f_n} - \mathbf{K}_{f_n\mathbf{u}}\mathbf{K}_{\mathbf{uu}}^{-1}\hat{\mathbf{m}}_{\mathbf{u}}. \tag{11}$$

Recall that for a twice-differentiable scalar function $h$

$$\nabla_{\boldsymbol{\Sigma}}\mathbb{E}_{\mathcal{N}(\mathbf{u};\,\boldsymbol{\mu},\,\boldsymbol{\Sigma})}\left[h(\mathbf{u})\right] = \mathbb{E}_{\mathcal{N}(\mathbf{u};\,\boldsymbol{\mu},\,\boldsymbol{\Sigma})}\left[H_h(\mathbf{u})\right] \tag{12}$$

where $H_h(\mathbf{u})$ is the Hessian of $h$ at $\mathbf{u}$. Thus, the gradient of the ELBO with respect to $\hat{\mathbf{K}}_{\mathbf{uu}}$ can be rewritten as

$$\nabla_{\hat{\mathbf{K}}_{\mathbf{uu}}}\mathcal{L}_{\text{ELBO}} = \sum_{n=1}^{N}\mathbb{E}_{\mathcal{N}\left(\mathbf{u};\,\hat{\mathbf{m}}_{\mathbf{u}},\,\hat{\mathbf{K}}_{\mathbf{uu}}\right)}\left[H_{h_n}(\mathbf{u})\right] - \frac{1}{2}\mathbf{K}_{\mathbf{uu}} + \frac{1}{2}\hat{\mathbf{K}}_{\mathbf{uu}} \tag{13}$$

where $h_n(\mathbf{u}) = \mathbb{E}_{\mathcal{N}\left(f_n;\,\mathbf{A}_n\mathbf{u}+a_n,\,\mathbf{K}_{f_n|\mathbf{u}}\right)}\left[\log p(\mathbf{y}_n|f_n\right].$

To determine an expression for $H_{h_n}$, we first consider the gradients of $h_n$. Let

$$\alpha_n(\beta_n) = \mathbb{E}_{\mathcal{N}\left(f_n;\,\beta_n,\,\mathbf{K}_{f_n|\mathbf{u}}\right)}\left[\log p(\mathbf{y}_n|f_n)\right] \tag{14}$$

$$\beta_n(\mathbf{u}) = \mathbf{A}_n\mathbf{u} + a_n. \tag{15}$$

The partial derivative of $h_n$ with respect to the $j^{\text{th}}$ element of $\mathbf{u}$ can be expressed as

$$\frac{\partial h_n}{\partial u_j}(\mathbf{u}) = \frac{\partial \alpha_n}{\partial \beta_n}(\beta_n(\mathbf{u}))\frac{\partial \beta_n}{\partial u_j}(\mathbf{u}). \tag{16}$$

Taking derivatives with respect to the $i^{\text{th}}$ element of $\mathbf{u}$ gives

$$\frac{\partial^2 h_n}{\partial u_j \partial u_i}(\mathbf{u}) = \frac{\partial^2 \alpha_n}{\partial \beta_n^2}(\beta_n(\mathbf{u}))\frac{\partial \beta_n}{\partial u_j}(\mathbf{u})\frac{\partial \beta_n}{\partial u_i}(\mathbf{u}) + \frac{\partial \alpha_n}{\partial \beta_n}(\beta_n(\mathbf{u}))\frac{\partial^2 \beta_n}{\partial u_j \partial u_i}(\mathbf{u}). \tag{17}$$

Thus, the Hessian is given by

$$H_{h_n}(\mathbf{u}) = \underbrace{\frac{\partial^2 \alpha_n}{\partial \beta_n^2}(\beta_n(\mathbf{u}))}_{\mathbb{R}}\underbrace{\nabla\beta_n(\mathbf{u})}_{N\times 1}\underbrace{\left[\nabla\beta_n(\mathbf{u})\right]^T}_{1\times N} + \underbrace{\frac{\partial \alpha_n}{\partial \beta_n}(\beta_n(\mathbf{u}))}_{\mathbb{R}}\underbrace{H_{\beta_n}(\mathbf{u})}_{N\times N}. \tag{18}$$

Since $\beta_n(\mathbf{u}) = \mathbf{A}_n\mathbf{u} + a_n$, we have $\nabla\beta_n(\mathbf{u}) = \mathbf{A}_n$ and $H_{\beta_n}(\mathbf{u}) = \mathbf{0}$. This allows us to write $\nabla_{\hat{\mathbf{K}}_{\mathbf{uu}}}\mathcal{L}_{\text{ELBO}}$ as

$$\nabla_{\hat{\mathbf{K}}_{\mathbf{uu}}}\mathcal{L}_{\text{ELBO}} = \sum_{n=1}^{N}\mathbb{E}_{\mathcal{N}\left(\mathbf{u};\,\hat{\mathbf{m}}_{\mathbf{u}},\,\hat{\mathbf{K}}_{\mathbf{uu}}\right)}\left[\frac{\partial^2 \alpha_n}{\partial \beta_n^2}(\beta_n(\mathbf{u}))\right]\mathbf{A}_n\mathbf{A}_n^T - \frac{1}{2}\mathbf{K}_{\mathbf{uu}} + \frac{1}{2}\hat{\mathbf{K}}_{\mathbf{uu}}. \tag{19}$$

The optimal covariance therefore satisfies

$$\hat{\mathbf{K}}_{\mathbf{uu}}^{-1} = \mathbf{K}_{\mathbf{uu}}^{-1} - 2\sum_{n=1}^{N}\mathbb{E}_{\mathcal{N}\left(\mathbf{u};\,\hat{\mathbf{m}}_{\mathbf{u}},\,\hat{\mathbf{K}}_{\mathbf{uu}}\right)}\left[\frac{\partial^2 \alpha_n}{\partial \beta_n^2}(\beta_n(\mathbf{u}))\right]\mathbf{A}_n\mathbf{A}_n^T. \tag{20}$$

Similarly, the gradient of the ELBO with respect to $\hat{\mathbf{m}}_{\mathbf{u}}$ can be written as

$$\nabla_{\hat{\mathbf{m}}_{\mathbf{u}}}\mathcal{L}_{\text{ELBO}} = \sum_{n=1}^{N}\nabla_{\hat{\mathbf{m}}_{\mathbf{u}}}\mathbb{E}_{\mathcal{N}\left(\mathbf{u};\,\hat{\mathbf{m}}_{\mathbf{u}},\,\hat{\mathbf{K}}_{\mathbf{uu}}\right)}\left[h_n(\mathbf{u})\right] - \mathbf{K}_{\mathbf{uu}}^{-1}(\hat{\mathbf{m}}_{\mathbf{u}} - \mathbf{m}_{\mathbf{u}})$$

$$= \sum_{n=1}^{N}\mathbb{E}_{\mathcal{N}\left(\mathbf{u};\,\hat{\mathbf{m}}_{\mathbf{u}},\,\hat{\mathbf{K}}_{\mathbf{uu}}\right)}\left[\nabla h_n(\mathbf{u})\right] - \mathbf{K}_{\mathbf{uu}}^{-1}(\hat{\mathbf{m}}_{\mathbf{u}} - \mathbf{m}_{\mathbf{u}}) \tag{21}$$

where we have used the fact that for a differentiable scalar function $h$

$$\nabla_{\boldsymbol{\mu}}\mathbb{E}_{\mathcal{N}(\mathbf{u};\,\boldsymbol{\mu},\,\boldsymbol{\Sigma})}\left[g(\mathbf{u})\right] = \mathbb{E}_{\mathcal{N}(\mathbf{u};\,\boldsymbol{\mu},\,\boldsymbol{\Sigma})}\left[\nabla g(\mathbf{u})\right]. \tag{22}$$

Using equation 16 and $\beta_n(\mathbf{u}) = \mathbf{A}_n\mathbf{u} + a_n$, we get

$$\nabla h_n(\mathbf{u}) = \frac{\partial \alpha_n}{\partial \beta_n}(\beta_n(\mathbf{u}))\mathbf{A}_n \tag{23}$$

giving

$$\nabla_{\hat{\mathbf{m}}_\mathbf{u}}\mathcal{L}_{\text{ELBO}} = \sum_{n=1}^{N} \mathbb{E}_{\mathcal{N}(\mathbf{u};\ \hat{\mathbf{m}}_\mathbf{u},\ \hat{\mathbf{K}}_{\mathbf{uu}})}\left[\frac{\partial \alpha_n}{\partial \beta_n}(\beta_n(\mathbf{u}))\right] - \mathbf{K}_{\mathbf{uu}}^{-1}(\hat{\mathbf{m}}_\mathbf{u} - \mathbf{m}_\mathbf{u}). \tag{24}$$

The optimal mean is therefore

$$\hat{\mathbf{m}}_\mathbf{u} = \mathbf{m}_\mathbf{u} - \sum_{n=1}^{N} \mathbb{E}_{\mathcal{N}(\mathbf{u};\ \hat{\mathbf{m}}_\mathbf{u},\ \hat{\mathbf{K}}_{\mathbf{uu}})}\left[\frac{\partial \alpha_n}{\partial \beta_n}(\beta_n(\mathbf{u}))\right]\mathbf{K}_{\mathbf{uu}}\mathbf{A}_n. \tag{25}$$

Equation 20 and equation 25 show that each $n^{\text{th}}$ observation contributes only a rank-1 term to the optimal approximate posterior precision matrix, corresponding to an optimum approximate posterior of the form

$$q(f) \propto p(f) \prod_{n=1}^{N} \mathcal{N}\left(\mathbf{K}_{f_n\mathbf{u}}\mathbf{K}_{\mathbf{uu}}^{-1}\mathbf{u};\ g_n,\ v_n\right) \tag{26}$$

where

$$g_n = -\mathbb{E}_{\mathcal{N}(\mathbf{u};\ \hat{\mathbf{m}}_\mathbf{u},\ \hat{\mathbf{K}}_{\mathbf{uu}})}\left[\frac{\partial \alpha_n}{\partial \beta_n}(\beta_n(\mathbf{u}))\right]v_n\hat{\mathbf{K}}_{\mathbf{uu}}^{-1}\mathbf{K}_{\mathbf{uu}} + \mathbf{A}_n^T\mathbf{m}_\mathbf{u} \tag{27}$$

$$1/v_n = -2\mathbb{E}_{\mathcal{N}(\mathbf{u};\ \hat{\mathbf{m}}_\mathbf{u},\ \hat{\mathbf{K}}_{\mathbf{uu}})}\left[\frac{\partial^2 \alpha_n}{\partial \beta_n^2}(\beta_n(\mathbf{u}))\right]. \tag{28}$$

For general likelihoods, these expressions cannot be solved exactly so $g_n$ and $v_n$ are freely optimised as variational parameters. When $N = M$, the inducing points are located at the observations and $\mathbf{A}_n\mathbf{A}_n^T$ is zero everywhere except for the $n^{\text{th}}$ element of its diagonal we recover the result of Opper & Archambeau (2009). Note the key role of the linearity of each $\beta_n$ in this result - without it $H_{\beta_n}$ would not necessarily be zero everywhere and the contribution of each $n^{\text{th}}$ term could have arbitrary rank.

## A.2 POSTERIOR GAUSSIAN PROCESS

For the sake of notational convenience, we shall assume $K = 1$. First, the mean and covariance of $q(\mathbf{u}) = \mathcal{N}\left(\mathbf{u};\ \hat{\mathbf{m}}_\mathbf{u},\ \hat{\mathbf{K}}_{\mathbf{uu}}\right) \propto p_{\theta_1}(\mathbf{u})\prod_{n=1}^{N_t} l_{\phi_l}(\mathbf{u}; \mathbf{y}_n^o, \mathbf{x}_n, \mathbf{Z})$ are given by

$$\hat{\mathbf{m}}_\mathbf{u} = \mathbf{K}_{\mathbf{uu}}\mathbf{\Phi}\mathbf{K}_{\mathbf{uf}}\mathbf{\Sigma}_{\phi_l,k}^{-1}\boldsymbol{\mu}_{\phi_l}$$
$$\hat{\mathbf{K}}_{\mathbf{uu}} = \mathbf{K}_{\mathbf{uu}}\mathbf{\Phi}\mathbf{K}_{\mathbf{uu}} \tag{29}$$

where $\mathbf{\Phi}^{-1} = \mathbf{K}_{\mathbf{uu}} + \mathbf{K}_{\mathbf{uf}}\mathbf{\Sigma}_{\phi_l}^{-1}\mathbf{K}_{\mathbf{fu}}$. The approximate posterior over some latent function value $f_*$ is obtained by marginalisation of the joint distribution:

$$q(f_*) = \int p_{\theta_1}(f_*|\mathbf{u})q(\mathbf{u})d\mathbf{u}$$
$$= \int \mathcal{N}\left(f_*;\ k_{f_*\mathbf{u}}\mathbf{K}_{\mathbf{uu}}^{-1}\mathbf{u},\ k_{f_*f_*} - k_{f_*\mathbf{u}}\mathbf{K}_{\mathbf{uu}}^{-1}k_{\mathbf{u}f_*}\right)\mathcal{N}\left(\mathbf{u};\ \hat{\mathbf{m}}_\mathbf{u},\ \hat{\mathbf{K}}_{\mathbf{uu}}\right)d\mathbf{u} \tag{30}$$
$$= \mathcal{N}\left(f_*;\ k_{f_*\mathbf{u}}\mathbf{K}_{\mathbf{uu}}^{-1}\hat{\mathbf{m}}_\mathbf{u},\ k_{f_*f_*} - k_{f_*\mathbf{u}}\mathbf{K}_{\mathbf{uu}}^{-1}k_{\mathbf{u}f_*} + k_{f_*\mathbf{u}}\mathbf{K}_{\mathbf{uu}}^{-1}\hat{\mathbf{K}}_{\mathbf{uu}}\mathbf{K}_{\mathbf{uu}}^{-1}k_{\mathbf{u}f_*}\right)$$

Substituting in equation 29 results in a mean and covariance function of the form

$$\hat{m}(\mathbf{x}) = k_{f\mathbf{u}}\mathbf{K}_{\mathbf{uu}}^{-1}\mathbf{\Phi}\mathbf{K}_{\mathbf{uf}}\mathbf{\Sigma}_{\phi_l,k}^{-1}\boldsymbol{\mu}_{\phi_l}$$
$$\hat{k}(\mathbf{x}, \mathbf{x}') = k_{ff'} - k_{f\mathbf{u}}\mathbf{K}_{\mathbf{uu}}^{-1}k_{\mathbf{u}f'} + k_{f\mathbf{u}}\mathbf{\Phi}k_{\mathbf{u}f'}. \tag{31}$$

## B  THE GP-VAE

As discuss in Section 3, the GP-VAE is described by the structured approximate posterior

$$q(f) = \frac{1}{\mathcal{Z}_q(\theta, \phi)} p_{\theta_1}(f) \prod_{n=1}^{N} l_{\phi_l}(\mathbf{f}_n; \mathbf{y}_n^o), \tag{32}$$

where $l_{\phi_l}(\mathbf{f}_n; \mathbf{y}_n^o) = \prod_{k=1}^{K} \mathcal{N}\left(\mathbf{f}_n;\ \boldsymbol{\mu}_{\phi_l}(\mathbf{y}_n^o),\ \text{diag}\ \boldsymbol{\sigma}_{\phi_l}^2(\mathbf{y}_n^o)\right)$, and corresponding ELBO

$$\mathcal{L}_{\text{ELBO}} = \mathbb{E}_{q(f)}\left[\log \frac{p_{\theta_1}(f) p_{\theta_2}(\mathbf{y}|\mathbf{f})}{\frac{1}{\mathcal{Z}_q(\theta,\phi)} p_{\theta_1}(f) l_{\phi_l}(\mathbf{f}; \mathbf{y})}\right] = \mathbb{E}_{q(\mathbf{f})}\left[\log \frac{p_{\theta_2}(\mathbf{y}|\mathbf{f})}{l_{\phi_l}(\mathbf{f}; \mathbf{y})}\right] + \log \mathcal{Z}_q(\theta, \phi). \tag{33}$$

### B.1  TRAINING THE GP-VAE

The final term in equation 33 has the closed-form expression

$$\mathcal{Z}_q(\theta, \phi) = \prod_{k=1}^{K} \sum_{k=1}^{K} \underbrace{\log \mathcal{N}\left(\boldsymbol{\mu}_{\phi_l,k};\ \mathbf{0},\ \mathbf{K}_{\mathbf{f}_k \mathbf{f}_k} + \boldsymbol{\Sigma}_{\phi_l,k}\right)}_{\log \mathcal{Z}_{q_k}(\theta, \phi)}. \tag{34}$$

which can be derived by noting that each $\mathcal{Z}_{q_k}(\theta, \phi)$ corresponds to the convolution between two multivariate Gaussians:

$$\mathcal{Z}_{q_k}(\theta, \phi) = \int \mathcal{N}\left(\mathbf{f}_k;\ \mathbf{0},\ \mathbf{K}_{\mathbf{f}_k \mathbf{f}_k}\right) \mathcal{N}\left(\boldsymbol{\mu}_{\phi_l,k} - \mathbf{f}_k;\ \mathbf{0},\ \boldsymbol{\Sigma}_{\phi_l,k}\right) d\mathbf{f}_k. \tag{35}$$

Similarly, a closed-form expression for $\mathbb{E}_{q(\mathbf{f})}\left[l_{\phi_l}(\mathbf{f}; \mathbf{y})\right]$ exists:

$$
\begin{aligned}
\mathbb{E}_{q(\mathbf{f})}\left[\log l_{\phi_l}(\mathbf{f}; \mathbf{y})\right] &= \sum_{k=1}^{K} \sum_{n=1}^{N} \mathbb{E}_{q(f_{nk})}\left[\log l_{\phi_l}(f_{nk}; \mathbf{y}_n^o)\right] \\
&= \sum_{k=1}^{K} \sum_{n=1}^{N} \mathbb{E}_{q(f_{nk})}\left[-\frac{(f_{nk} - \mu_{\phi_l,k}(\mathbf{y}_n^o))^2}{2\sigma_{\phi_l,k}^2(\mathbf{y}_n^o)} - \frac{1}{2}\log|2\pi\sigma_{\phi_l,k}^2(\mathbf{y}_n^o)|\right] \\
&= \sum_{k=1}^{K} \sum_{n=1}^{N} -\frac{\left[\hat{\boldsymbol{\Sigma}}_k\right]_{nn} + (\hat{\mu}_{k,n} - \mu_{\phi_l,k}(\mathbf{y}_n^o))^2}{2\sigma_{\phi_l,k}^2(\mathbf{y}_n^o)} - \frac{1}{2}\log|2\pi\sigma_{\phi_l,k}^2(\mathbf{y}_n^o)| \\
&= \sum_{k=1}^{K} \sum_{n=1}^{N} \log \mathcal{N}\left(\hat{\mu}_{k,n};\ \mu_{\phi_l,k}(\mathbf{y}_n^o),\ \sigma_{\phi_l,k}^2(\mathbf{y}_n^o)\right) - \frac{\left[\hat{\boldsymbol{\Sigma}}_k\right]_{nn}}{2\sigma_{\phi_l,k}^2(\mathbf{y}_n^o)} \\
&= \sum_{k=1}^{K} \log \mathcal{N}\left(\hat{\boldsymbol{\mu}}_k;\ \boldsymbol{\mu}_{\phi_l,k},\ \boldsymbol{\Sigma}_{\phi_l,k}\right) - \sum_{n=1}^{N} \frac{\left[\hat{\boldsymbol{\Sigma}}_k\right]_{nn}}{2\sigma_{\phi_l,k}^2(\mathbf{y}_n)} \tag{36}
\end{aligned}
$$

where $\hat{\boldsymbol{\Sigma}}_k = \hat{k}_k\left(\mathbf{X}, \mathbf{X}'\right)$ and $\hat{\boldsymbol{\mu}}_k = \hat{m}_k(\mathbf{X})$, with

$$
\begin{aligned}
\hat{m}_k(\mathbf{x}) &= k_{f_k \mathbf{u}_k}\left(\mathbf{K}_{\mathbf{u}_k \mathbf{u}_k} + \boldsymbol{\Sigma}_{\phi_l,k}\right)^{-1} \boldsymbol{\mu}_{\phi_l,k} \\
\hat{k}_k(\mathbf{x}) &= k_{f_k f_k'} - k_{f_k \mathbf{u}_k}\left(\mathbf{K}_{\mathbf{u}_k \mathbf{u}_k} + \boldsymbol{\Sigma}_{\phi_l,k}\right)^{-1} k_{\mathbf{u}_k f_k}. \tag{37}
\end{aligned}
$$

$\mathbb{E}_{q(\mathbf{f})}\left[\log p_{\theta_2}(\mathbf{y}|\mathbf{f})\right]$ is intractable, hence must be approximated by a Monte Carlo estimate. Together with the closed-form expressions for the other two terms we can form an unbiased estimate of the ELBO, the gradients of which can be estimated using the reparameterisation trick (Kingma & Welling, 2014).

## B.2  AN ALTERNATIVE SPARSE APPROXIMATION

An alternative approach to introducing a sparse GP approximation is directly parameterise the structured approximate posterior at inducing points $\mathbf{u}$:

$$q(f) = \frac{1}{\mathcal{Z}_q(\theta, \phi)} p_{\theta_1}(f) \prod_{n=1}^{N} l_{\phi_l}(\mathbf{u}; \mathbf{y}_n^o, \mathbf{x}_n, \mathbf{Z}) \tag{38}$$

where $l_{\phi_l}(\mathbf{u}; \mathbf{y}_n^o, \mathbf{x}_n, \mathbf{Z})$, the approximate likelihood, is a fully-factorised Gaussian distribution parameterised by a partial inference network:

$$l_{\phi_l}(\mathbf{u}; \mathbf{y}_n^o, \mathbf{x}_n, \mathbf{Z}) = \prod_{k=1}^{K} \prod_{m=1}^{M} \mathcal{N}\left(u_{mk}; \mu_{\phi,k}(\mathbf{y}_n^o), \sigma_{\phi,k}^2(\mathbf{y}_n^o)\right). \tag{39}$$

In general, each factor $l_{\phi_l}(u_{mk}; \mathbf{y}_n^o, \mathbf{z}_{mk}, \mathbf{x}_n)$ conditions on data at locations different to that of the inducing point. The strength of the dependence between these values is determined by the two input locations themselves. To account for this, we introduce the use of an inference network that, for each observation/inducing point pair $(u_{mk}, \mathbf{y}_n)$, maps from $(\mathbf{z}_{mk}, \mathbf{x}_n, \mathbf{y}_n^o)$ to parameters of the approximate likelihood factor.

Whilst this approach has the same first order computational complexity as that used by the SGP-VAE, having to making forward and backward passes through the inference network $KNM$ renders it significantly more computationally expensive for even moderately sized datasets. Whereas the approach adopted by the SGP-VAE employs an deterministic transformation of the outputs of the inference network based on the covariance function, this approach can be interpreted as learning an appropriate dependency between input locations. In practice, we found the use of this approach to result in worse predictive performance.

## C  MEMORY REQUIREMENTS

Assuming input locations and inducing point locations are shared across tasks, we require storing $\{\mathbf{K}_{\mathbf{u}_k \mathbf{f}_k^{(t)}} + \mathbf{K}_{\mathbf{u}_k \mathbf{u}_k}\}_{k=1}^{K}$ and $\mathbf{K}_{\mathbf{f}_k^{(t)} \mathbf{f}_k^{(t)}}$ in memory, which is $\mathcal{O}\left(KMN + KM^2 + N^2\right)$. For the SGP-VAE, we also require storing $\phi$ and instantiating $\{\boldsymbol{\mu}_{\phi_l,k}^{(t)}, \boldsymbol{\Sigma}_{\phi_l,k}^{(t)}\}_{k=1}^{K}$, which is $\mathcal{O}\left(|\phi_l| + KMD + 2KN\right)$. Collectively, this results in the memory requirement $\mathcal{O}\left(KNM + KM^2 + N^2 + |\phi_l| + KMD + 2KN\right)$.

If we were to employ the same sparse structured approximate posterior, but replace the output of the inference network with free-form variational parameters, the memory requirement is $\mathcal{O}\left(KNM + KM^2 + N^2 + KMD + 2TKN\right)$.[7] Alternatively, if we were to let $q(\mathbf{u})$ to be parameterised by free-form Cholesky factors and means, the memory requirement is $\mathcal{O}\left(KNM + KM^2 + N^2 + KMD + TKM(M+1)/2 + TKM\right)$. Table 3 compares the first order approximations. Importantly, the use of amortisation across tasks stops the memory scaling with the number of tasks.

**Table 3:** A comparison between the memory requirements of approximate posteriors.

| $q(\mathbf{u})$ | Amortised? | Memory requirement |
|---|---|---|
| $p(\mathbf{u}) \prod_n l_n(\mathbf{u})$ | Yes | $\mathcal{O}\left(KNM + KM^2 + N^2 + |\phi_l|\right)$ |
| $p(\mathbf{u}) \prod_n l_n(\mathbf{u})$ | No | $\mathcal{O}\left(KNM + KM^2 + N^2 + TKN\right)$ |
| $q(\mathbf{u})$ | No | $\mathcal{O}\left(KNM + TKM^2\right)$ |

## D  MULTI-OUTPUT GAUSSIAN PROCESSES

Through consideration of the interchange of input dependencies and likelihood functions, we can shed light on the relationship between the probabilistic model employed by the SGP-VAE and other multi-output GP models. These relationships are summarised in Figure 4.

---

[7]Note we only require evaluating a single $\mathbf{K}_{\mathbf{f}_k^{(t)} \mathbf{f}_k^{(t)}}$ at each update.

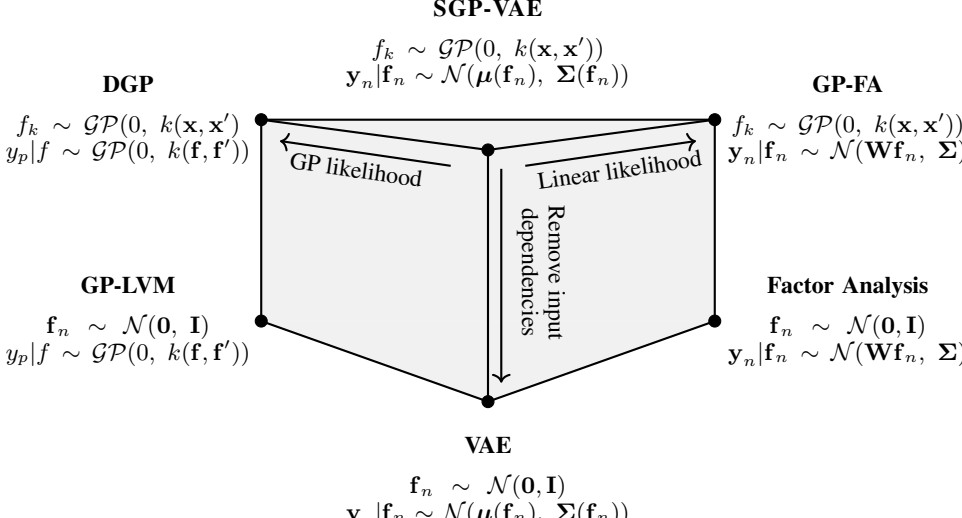

**Figure 4:** A unifying perspective on multi-output GPs.

**Linear Multi-Output Gaussian Processes** Replacing the likelihood with a linear likelihood function characterises a family of linear multi-output GPs, defined by a linear transformation of $K$ independent latent GPs:

$$f \sim \prod_{k=1}^{K} \mathcal{GP} \left(0, k_{\theta_{1,k}}(\mathbf{x}, \mathbf{x}')\right)$$

$$\mathbf{y}|f \sim \prod_{n=1}^{N} \mathcal{N} \left(\mathbf{y}_n;\ \mathbf{W}\mathbf{f}_n,\ \mathbf{\Sigma}\right). \tag{40}$$

The family includes Teh et al.'s (2005) semiparametric latent factor model, Yu et al.'s (2009) GP factor analysis (GP-FA) and Bonilla et al.'s (2008) class of multi-task GPs. Notably, removing input dependencies by choosing $k_{\theta_{1,k}}(\mathbf{x}, \mathbf{x}') = \delta(\mathbf{x}, \mathbf{x}')$ recovers factor analysis, or equivalently, probabilistic principal component analysis (Tipping & Bishop, 1999) when $\mathbf{\Sigma} = \sigma^2\mathbf{I}$. Akin to the relationship between factor analysis and linear multi-output GPs, the probabilistic model employed by standard VAEs can be viewed as a special, instantaneous case of the SGP-VAE's.

**Deep Gaussian Processes** Single hidden layer deep GPs (DGPs) (Damianou & Lawrence, 2013) are characterised by the use of a GP likelihood function, giving rise to the probabilistic model

$$f \sim \prod_{k=1}^{K} \mathcal{GP} \left(0, k_{\theta_{1,k}}(\mathbf{x}, \mathbf{x}')\right)$$

$$y|f \sim \prod_{p=1}^{P} \mathcal{GP} \left(0, k_{\theta_{2,p}}(f(\mathbf{x})f(\mathbf{x}'))\right) \tag{41}$$

where $\mathbf{y}_n = y(\mathbf{x}_n)$. The GP latent variable model (GP-LVM) (Lawrence & Moore, 2007) is the special, instantaneous case of single layered DGPs. Multi-layered DGPs are recovered using a hierarchical latent space with conditional GP priors between each layer.

## E   EXPERIMENTAL DETAILS

Whilst the theory outlined in Section 2 describes a general decoder parameterising both the mean and variance of the likelihood, we experienced difficulty training SGP-VAEs using a learnt variance, especially for high-dimensional observations. Thus, for the experiments detailed in this paper we use a shared variance across all observations. We use the Adam optimiser (Kingma & Ba, 2014) with a constant learning rate of 0.001. Unless stated otherwise, we estimate the gradients of the

ELBO using a single sample and the ELBO itself using 100 samples. The predictive distributions are approximated as Gaussian with means and variances estimated by propagating samples from $q(f)$ through the decoder. For each experiment, we normalise the observations using the means and standard deviations of the data in the training set.

The computational complexity of performing variational inference (VI) in the full GP-VAE, per update, is dominated by the $\mathcal{O}\left(KN^3\right)$ cost associated with inverting the set of $K$ $N \times N$ matrices, $\{\mathbf{K}_{\mathbf{f}_k\mathbf{f}_k} + \mathbf{\Sigma}_{\phi_l,k}\}_{k=1}^K$. This can quickly become burdensome for even moderately sized datasets. A pragmatic workaround is to use a biased estimate of the ELBO using $\tilde{N} < N$ data points:

$$\tilde{\mathcal{L}}_{\text{ELBO}}^{\tilde{N}} = \frac{N}{\tilde{N}} \left[ \mathbb{E}_{q(\tilde{\mathbf{f}})} \left[ \log \frac{p_{\theta_2}(\tilde{\mathbf{y}}|\tilde{\mathbf{f}})}{l_\phi(\tilde{\mathbf{f}}|\tilde{\mathbf{y}})} \right] + \log \tilde{\mathcal{Z}}_q(\theta, \phi) \right]. \tag{42}$$

$\tilde{\mathbf{y}}$ and $\tilde{\mathbf{f}}$ denote the mini-batch of $\tilde{N}$ observations and their corresponding latent variables, respectively. The bias is introduced due to the normalisation constant, which does not satisfy $\frac{N}{\tilde{N}}\mathbb{E}\left[\log \tilde{\mathcal{Z}}_q(\theta, \phi)\right] = \mathbb{E}\left[\log \mathcal{Z}_q(\theta, \phi)\right]$. Nevertheless, the mini-batch estimator will be a reasonable approximation to the full estimator provided the lengthscale of the GP prior is not too large.[8] Mini-batching cannot be used to reduce the $\mathcal{O}\left(KN^3\right)$ cost of performing inference at test time, hence sparse approximations are necessary for large datasets.

### E.1 SMALL-SCALE EEG

For all GP-VAE models, we use a three-dimensional latent space, each using squared exponential (SE) kernels with lengthscales and scales initialised to 0.1 and 1, respectively. All DNNs, except for those in PointNet and IndexNet, use two hidden layers of 20 units and ReLU activation functions. PointNet and IndexNet employ DNNs with a single hidden layer of 20 units and a 20-dimensional intermediate representation. Each model is trained for 3000 epochs using a batch size of 100, with the procedure repeated 15 times. Following (Requeima et al., 2019), the performance of each model is evaluated using the standardised mean squared error (SMSE) and negative log-likelihood (NLL). The mean $\pm$ standard deviation of the performance metrics for the 10 iterations with the highest ELBO is reported.[9]

### E.2 JURA

We use a two-dimensional latent space for all GP-VAE models with SE kernels with lengthscales and scales initialised to 1. This permits a fair comparison with other multi-output GP methods which also use two latent dimensions with SE kernels. For all DNNs except for those in IndexNet, we use two hidden layers of 20 units and ReLU activation functions. IndexNet uses DNNs with a single hidden layer of 20 units and a 20-dimensional intermediate representation. Following Goovaerts (1997) and Lawrence (2004), the performance of each model is evaluated using the mean absolute error (MAE) averaged across 10 different initialisations. The 10 different initialisations are identified from a body of 15 as those with the highest training set ELBO. For each initialisation the GP-VAE models are trained for 3000 epochs using a batch size of 100.

### E.3 LARGE-SCALE EEG

In both experiments, for each trial in the test set we simulate simultaneous electrode 'blackouts' by removing any 4 sample period at random with 25% probability. Additionally, we simulate individual electrode 'blackouts' by removing any 16 sample period from at random with 50% probability from the training set. For the first experiment, we also remove any 16 sample period at random with 50% probability from the test set. For the second experiment, we remove any 16 sample period at random with 10% probability. All models are trained for 100 epochs, with the procedure repeated five times, and use a 10-dimensional latent space with SE kernels and lengthscales initialised to 1

---

[8]In which case the off-diagonal terms in the covariance matrix will be large making the approximation $p_{\theta_1}(\mathbf{f}) = \prod p_{\theta_1}(\tilde{\mathbf{f}})$ extremely crude.

[9]We found that the GP-VAE occasionally got stuck in very poor local optima. Since the ELBO is calculated on the training set alone, the experimental procedure is still valid.

and 0.1, respectively. All DNNs, except for those in PointNet and IndexNet, use four hidden layers of 50 units and ReLU activation functions. PointNet and IndexNet employ DNNs with two hidden layers of 50 units and a 50-dimensional intermediate representation.

### E.4   BOUNCING BALL

To ensure a fair comparison with the SVAE and SIN, we adopt an identical architecture for the inference network and decoder in the original experiment. In particular, we use DNNs with two hidden layers of 50 units and hyperbolic tangent activation functions. Whilst both Johnson et al. and Lin et al. use eight-dimensional latent spaces, we consider a GP-VAE with a one-dimensional latent space and periodic GP kernel. For the more complex experiment, we use a SGP-VAE with fixed inducing points placed every 50 samples. We also increase the number of hidden units in each layer of the DNNs to 256 and use a two-dimensional latent space - one for each ball.

### E.5   WEATHER STATION

The spatial location of each weather station is determined by its latitude, longitude and elevation above sea level. The rates of missingness in the dataset vary, with 6.3%, 14.0%, 18.9%, 47.3% and 93.2% of values missing for each of the five weather variables, respectively. Alongside the average temperature for the middle five days, we simulate additional missingness from the test datasets by removing 25% of the minimum and maximum temperature values. Each model is trained on the data from 1980 using a single group per update for 50 epochs, with the performance evaluated on the data from both 1980 and 1981 using the root mean squared error (RMSE) and NLL averaged across five runs. We use a three-dimensional latent space with SE kernels and lengthscales initialised to 1. All DNNs, except for those in PointNet and IndexNet, use four hidden layers of 20 units and ReLU activation functions. PointNet and IndexNet employ DNNs with two hidden layers of 20 units and a 20-dimensional intermediate representation. Inducing point locations are initialised using k-means clustering, and are shared across latent dimensions and groups. The VAE uses FactorNet. We consider independent GPs modelling the seven point time series for each variable and each station, with model parameters shared across groups. No comparison to other sparse GP approaches is made and there is no existing framework for performing approximate inference in sparse GP models conditioned on previously unobserved data.

## F   FURTHER EXPERIMENTATION

### F.1   SMALL SCALE EXPERIMENTS

**Table 4:** A comparison between multi-output GP models on the EEG and Jura experiments.

|      | **Metric** | GP-VAE | GPPVAE |
|------|------------|--------|--------|
| EEG  | SMSE | **0.24 (0.02)** | 0.524 (0.11) |
|      | NLL | **2.01 (0.28)** | **1.92 (0.05)** |
| Jura | MAE | **0.40 (0.01)** | 0.49 (0.02) |
|      | NLL | **1.00 (0.06)** | **0.99 (0.04)** |

Table 4 compares the performance of the GP-VAE to that of the GPPVAE, In all cases, FactorNet is used to handle missing data. We emphasise that the GP-VAE and GPPVAE employ identical probabilistic models, with the only difference being the form of the approximate posterior. The superior predictive performance of the GP-VAE can therefore be accredited to the use of the structured approximate posterior as opposed to the mean-field approximate posterior used by the GPPVAE.

### F.2   SYNTHETIC BOUNCING BALL EXPERIMENT

The original dataset consists of 80 12-dimensional image sequences each of length 50, with the task being to predict the trajectory of the ball given a prefix of a longer sequence. The image sequences are generated at random by uniformly sampling the starting position of the ball whilst keeping the

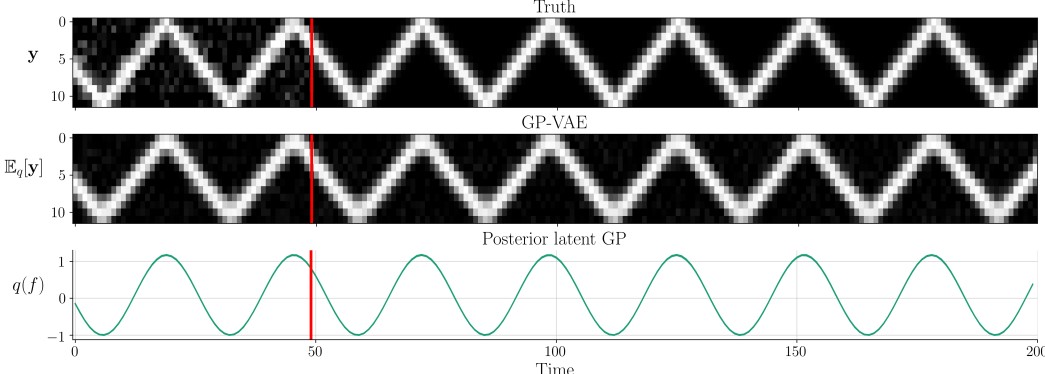

**Figure 5:** A comparison between the mean of the GP-VAE's posterior predictive distribution (middle) and the ground truth (top) conditioned on noisy observations up to the red line. The latent approximate GP posterior is also shown (bottom).

bouncing frequency fixed. Figure 5 compares the posterior latent GP and mean of the posterior predictive distribution with the ground truth for a single image sequence using just a single latent dimension. As demonstrated in the more more complex experiment, the GP-VAE is able to recover the ground truth with almost exact precision.

Following Lin et al. (2018), Figure 1a evaluates the $\tau$-steps ahead predictive performance of the GP-VAE using the mean absolute error, defined as

$$\sum_{n=1}^{N_{\text{test}}} \sum_{t=1}^{T-\tau} \frac{1}{N_{\text{test}}(T-\tau)d} \left\| \mathbf{y}^*_{n,t+\tau} - \mathbb{E}_{q(y_{n,t+\tau}|y_{n,1:t})} \left[ \mathbf{y}_{n,t+\tau} \right] \right\|_1 \tag{43}$$

where $N_{\text{test}}$ is the number of test image sequences with $T$ time steps and $\mathbf{y}^*_{n,t+\tau}$ denotes the noiseless observation at time step $t + \tau$.

## G  PARTIAL INFERENCE NETWORK COMPUTATIONAL GRAPHS

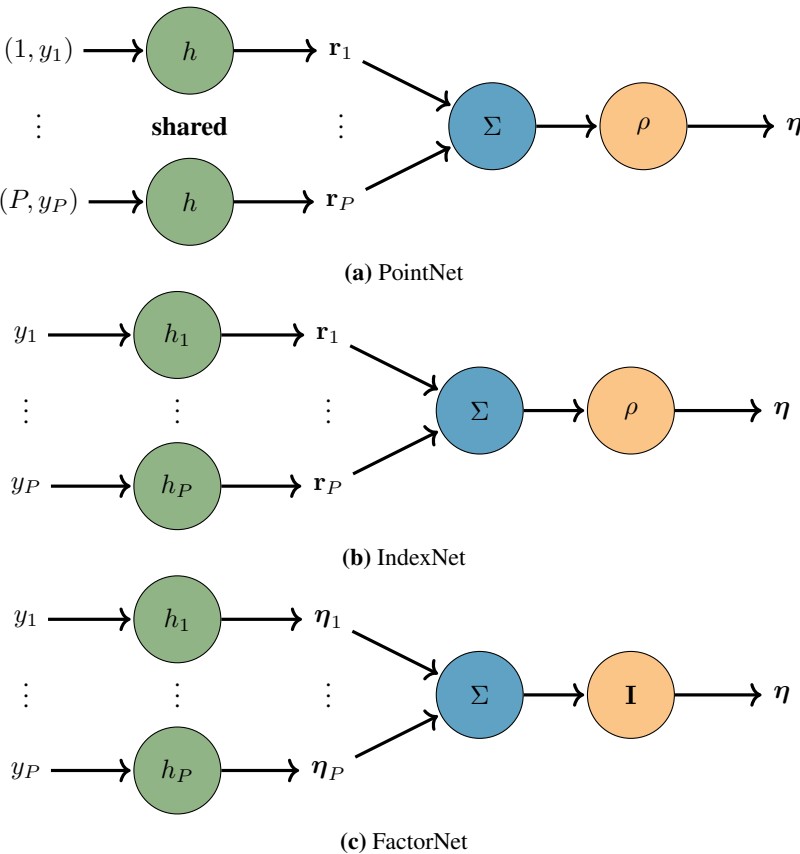

**Figure 6:** An illustration of the three different partial inference network specifications discuss in Section 2.4. $\boldsymbol{\eta}$ denotes the vector of natural parameters of the multi-variate Gaussian being parameterised.

