# OpenReview forum: "Sparse Gaussian Process Variational Autoencoders"
_ICLR.cc/2021/Conference — Reject_

### Official Review · AnonReviewer4 · 2020-10-25
**The paper proposes a multi-task multi-output GP, with sparse approximation and imputation of data, making use of inference networks.**

**Rating:** 6
**Confidence:** 3

**Review:**

The model proposed by the paper is a multi-task, multi-output GP, with likelihoods driven by a decoder network. While this model is not new --- same as that of Pearce 2020 as acknowledged by the authors the related work section --- this paper includes the sparse approximation and also handles missing data.

The paper can be made clearer:
1. The paper introduces "partial inference network" in section 2.1. Since this plays a central part in the contribution of the paper, I hope the authors can expand on this. For example, in what way is it "partial" and how is it different from normal inference networks, and what are the challenges. From a GP and Bayesian inference perspective, missing data is no problem, as acknowledge by the authors in section 2.3. Hence, why is it that the normal mechanism fails here? Although section 2.3 is section dedicated to this, it is insufficient.

2. N and P in the last para of section 2.1 are not defined.

3. The first and third shortcomings of sparse GP frameworks claimed by the authors in the last three sentences of section 2.1 do not really exist in existing sparse approximations to GP. Some clarification is needed here.

4. Some figures accompanying the different approaches in section 2.3 will be very helpful.

5. The section title for section 2 is "spatial-temporal". It is good to lay out what is the spatial and what is the temporal aspect for each of the data sets in section 4. For example, I do not really find the "temporal" dimension in Jura. One way around this is to have a more general section title.

The paper is contributes to the community and the work is correct, so *accept*. It is *not a stronger accept* because I find the "sparse" contribution minimal given the vast literature of sparse approximations to GP; and I find the "missing-data" contribution confusing.

---

> ### Author Response · Authors · 2020-11-18
> **Response to review #3 (part 1/2)**
>
> Thank you for taking the time to read our paper in detail, and for providing helpful comments. We have included a section (2.1) in the revised version of the paper which elaborates on the significance of our contribution. Further, we have provided a detailed discussion as to how our method rectifies three major limitations of current sparse GP approximations at the end of section 2.2. Below, we address each of your comments individually.
>
> **"The paper introduces "partial inference network" in section 2.1. Since this plays a central part in the contribution of the paper, I hope the authors can expand on this. For example, in what way is it "partial" and how is it different from normal inference networks, and what are the challenges. From a GP and Bayesian inference perspective, missing data is no problem, as acknowledge by the authors in section 2.3. Hence, why is it that the normal mechanism fails here? Although section 2.3 is section dedicated to this, it is insufficient."**
> A characteristic common of spatio-temporal datasets is the presence of missing data. In general, amortised inference is achieved using a mapping from observations to parameters of an approximate posterior. This mapping is typically achieved using a neural network (NN), whose first layer is a linear transformation of observations. When the observation contains missing values, there is nothing to apply this linear transformation to and so the standard architecture cannot be used.
>
> A common workaround is retain the standard inference network architecture whilst imputing these missing values with zeros. This is theoretically unappealing, however, as the inference network can no longer distinguish between a true zero and a missing value. An elegant workaround, first proposed by Ma et al. (2018) in the partial VAE, is to reinterpret partial observations as a set and use permutation invariant set functions as the inference network—this is exactly what partial inference networks are. They are different to standard inference networks in that they operate on sets of arbitrary size, as opposed to fixed dimensional vectors. Partial observations are not fixed dimensional vectors—the number of observations varies in size—thus the use of partial inference networks is a necessity.
>
> **"N and P in the last para of section 2.1 are not defined."**
> P is defined as the dimension of the observations at the beginning of section 2. The reviewer is correct in stating that $N$ is not defined --- we apologise for this. We have made the appropriate correction in the revised version of the paper and now only refer to $N^{(t)}$, which is the number of data points in task $t$.
>
> **"The first and third shortcomings of sparse GP frameworks claimed by the authors in the last three sentences of section 2.1 do not really exist in existing sparse approximations to GP. Some clarification is needed here."**
> The first claim made in the paper is that "...the inference network can be used to condition on previously unobserved data without needing to learn new variational parameters". Suppose we use the standard sparse GP variational approximation $q(f) = p(f|u)q(u)$ where $q(u) = \mathcal{N}(u; m, S)$ with $m$, $S$ and $Z$ (the inducing point locations) being our variational parameters, which are found by maximising the ELBO. When more data are observed, the optimum $m$ and $S$ will almost certainly have changed, and thus the ELBO needs to be re-optimised. When we use amortisation to parameterise $q(u)$, the approximate posterior is `automatically' updated by mapping from the new observations to their corresponding approximate likelihood terms, and including these approximate likelihood terms in the approximate posterior. Thus, the approximate posterior is updated due to new observations without having to re-optimise the ELBO.
>
> The third claim is that "...if the inducing point locations are fixed, then the number of variational parameters does not depend on the size of the dataset, even as more inducing points are added.". For the standard sparse GP variational approximation, as inducing points are added $m$ and $S$ increase in size, corresponding to the mean and covariance values at the new inducing point locations. These are new variational parameters, which must be optimised by maximising the ELBO. When amortisation is used, $m$ and $S$ are no longer `free-form' variational parameters—they are simply the product of the output of the inference network (whose parameters are the variational parameters). The inference network does not have to change as more inducing points are added, thus no new variational parameters are introduced.
>
> We are very happy to expand the existing discussion of these points in the paper to make them clearer.
>
> **"Some figures accompanying the different approaches in section 2.3 will be very helpful."**
> We agree! In fact the paper already includes Appendix G which contains diagrams showing the computational graph for each partial inference network.

---

> > ### Author Response · Authors · 2020-11-18
> > **Response to review #3 (part 2/2)**
> >
> > **"The section title for section 2 is "spatial-temporal". It is good to lay out what is the spatial and what is the temporal aspect for each of the data sets in section 4. For example, I do not really find the "temporal" dimension in Jura. One way around this is to have a more general section title."**
> > We use the general term 'spatio-temporal' to encompass to datasets that are distributed across either space or time, or both. The reviewer is correct in stating that the Jura dataset is not distributed across time, however it is distributed across space.
> >
> > [Ma et al., 2018] EDDI: Efficient dynamic discovery of high-value information with partial VAE.

---

### Official Review · AnonReviewer2 · 2020-10-29
**Novelty and comparative study should be highlighted**

**Rating:** 6
**Confidence:** 3

**Review:**

This paper attempts to enhance the inference in multi-output GPs on previously unobserved data using amortised variational inference. This model is proposed to overcome the shortcomings in current GP-DGMs. The major concerns regarding this paper are summarized as below.

One of the two main contributions in this paper is introducing sparse GP via amortized variational inference in order to replace the time-consuming GP, the novelty of which however seems to be incremental.

The modeling of task dependency should be highlighted. In my understanding, this paper employs a MLP to mix K latent functions in a nonlinear manner through equation 1 for each output. The NN parameters are shared across tasks, which however might be a too strong assumption and thus could deteriorate the performance of multi-task learning.

The comparative results in Table 1 cannot showcase the superiority of GP-VAE in comparison to the state-of-the-art multi-output GP. It performs worse than GPAR in terms of NLL on the EEG dataset, and is comparable to GPAR on the two datasets in terms of SMSE.

---

> ### Author Response · Authors · 2020-11-18
> **Response to review #2**
>
> We greatly appreciate the reviewers feedback, and have revised the submission to include a more thorough explanation of our main contributions. We address the reviewers comments individually below.
>
> **"One of the two main contributions in this paper is introducing sparse GP via amortized variational inference in order to replace the time-consuming GP, the novelty of which however seems to be incremental."**
> The combination of amortisation and sparse GP approximations is not straightforward. Consider the standard form of the sparse GP approximate posterior $q(f) = p(f|u)q(u)$ qith $q(u) = \mathcal{N}(u; m, S)$ where $m$ and $S$ are our variational parameters. This does not decompose into a product over $N$ factors and is therefore not amendable to per-datapoint amortisation. In Appendix 1, we demonstrate that the optimum $q(u)$ can be expressed as the product of the prior $p(u)$ multiplied by the product of $N$ factors, one for each datapoint. This naturally lends itself to per-datapoint amortisation by using an inference network to map from each observation to the parameters of the approximate likelihood. Further, this result is general: optimality holds for any GP model with arbitrary likelihood and independent GP priors.
>
> The combination of amortisation and sparse approximations addresses three major shortcomings of existing sparse approximations—we discuss this in detail in the revised version of the paper.
>
> **"The modeling of task dependency should be highlighted. In my understanding, this paper employs a MLP to mix K latent functions in a nonlinear manner through equation 1 for each output. The NN parameters are shared across tasks, which however might be a too strong assumption and thus could deteriorate the performance of multi-task learning."**
> We thank the reviewer for raising a valid concern regarding the sharing of NN parameters across tasks. This concern is equivalent to that of sharing the model parameters across tasks, the validity of which hinges on the degree of similarity between observations across tasks. For all datasets we considered, we assumed the variation in observations between tasks to be due to the variation in the underlying 'state of the world' which we modelled by performing inference using the latent GPs. For settings in which the variation between tasks is significant, the complexity of the latent space (i.e. number of latent GPs) can be increased such this variation can be captured.
>
> **"The comparative results in Table 1 cannot showcase the superiority of GP-VAE in comparison to the state-of-the-art multi-output GP. It performs worse than GPAR in terms of NLL on the EEG dataset, and is comparable to GPAR on the two datasets in terms of SMSE."**
> GPAR represents the current state-of-the-art for multi-output GP models on both the EEG and Jura experiments by a significant margin. The reviewer is correct in stating that the SGP-VAE doesn't offer a big advantage on these small datasets—our intention was to demonstrate the utility of the SGP-VAE on small datasets, and ensure that amortisation, especially in the presence of partially observed data, is not at the expense of predictive performance. The fact that we achieved comparable to state-of-the-art performance confirms this.
>
> Whilst GPAR performs well in the small datasets above, it has two key limitations which severely restrict the range of problems on which it can be deployed. First, it can only be used with specific patterns of missing data and cannot be used when the missingness pattern is arbitrary. Second, it is not scalable and would require further development to handle the large datasets considered in this paper. In contrast, the SGP-VAE is far more flexible: it handles arbitrary patterns of missingness, and scales to large numbers of datapoints and tasks. Another advantage of the SGP-VAE is that is models $P$ outputs with just $K$ latent GPs. This differs to GPAR, which uses $P$ GPs. Whilst this isn't an issue for the small-scale experiments, it quickly becomes computationally burdensome when $P$ becomes large (e.g. the bouncing ball experiment would demand GPAR to use 100 GPs, whereas the SGP-VAE uses just 2). The true efficacy of the SGP-VAE is showcased in the final two experiments, where the number of datapoints and tasks is large, and the patterns of missingness are random.

---

### Official Review · AnonReviewer1 · 2020-10-29
**Interesting encoder, missing some  empirical results.**

**Rating:** 6
**Confidence:** 4

**Review:**

In this work generative models using a GP as prior and a deep network as likelihood (GP-DGMs) are considered. In the VAE formalism for inference, the novelty of this paper is located in the encoder: It is sparse and the posterior can be computed even when part of the observations are missing. Sparsity is obtained using inducing inputs and the missing observations are handled through the use of deep sets, i.e. the observations aren't given as a vector, but as a permutation-invariant set of (index, value) pairs.

The idea is  interesting and well executed, and the experimental results are certainly good when compared with the provided baselines. However, there are many unclear details in the experiments. I was left with quite a few unanswered questions:
- In the cases in which no imputation is needed, how does the proposed SGP-VAE compare with a standard encoder (instead of one based on deep sets). I.e., are we paying a price for the ability to handle missing data, or are we getting even better results than with a traditional encoder?
- How does the model compare with prior similar work, such as the GPPVAE? In principle, it should be easy to improve on this baseline.
- How good is the model itself? Here we see results based on amortized inference, but what if we weren't using it? Even though this case would be slow and maybe not able to handle the entire dataset, this can be done in a subset. In general, it is unclear how much is being lost by a) amortized inference; b) sparsity; c) use of deep sets (in this latter case, it could that something is won instead).
- In the Japanese Weather experiment, when the standard VAE is used (with no mean imputation), how are the unobserved values being dealt with?
- In the Japanese Weather experiment, the baselines don't seem very strong. A standard VAE and independent GPs don't seem to be particularly suited to this spatiotemporal weather prediction.
- In general, I'm not sure where the advantage seen in these experiments is coming from. For instance, when compared with published results on the same data (e.g., GPAR on EEG), the SGP-VAE doesn't seem to offer a big advantage. In the other experiments, no strong baselines for that specific  data seem to be provided.
- I would be curious to know in which of these datasets the model itself is superior (if we were to skip the encoder and use, say, MCMC for inference) and in which ones it isn't.

Thanks for your response, clarifying.

---

> ### Author Response · Authors · 2020-11-18
> **Response to review #1 (part 1/2)**
>
> Many thanks for taking the time to read our paper thoroughly and providing a detailed assessment. We found your comments very helpful, and have incorporated your suggestions into the revised version. Below, we address each of your comments individually.
>
> **"In the cases in which no imputation is needed, how does the proposed SGP-VAE compare with a standard encoder (instead of one based on deep sets). I.e., are we paying a price for the ability to handle missing data, or are we getting even better results than with a traditional encoder?"**
> All experiments considered in this paper contained missing data, thus the ability to handle partial observation is a necessity. These experiments therefore do not shed light on how the SGP-VAE performs when using a standard encoder. Nevertheless, it is instructive to compare the use of zero imputation---for which the standard encoder was used---to that of the partial inference networks. For the EEG and Jura experiments, we found that their was no significant difference between the performance of the SGP-VAE using zero imputation and FactorNet. However, for the Japanese weather experiment we found the use of zero imputation (and hence standard encoder) performed substantially worse than the use of IndexNet or FactorNet.
>
> **"How does the model compare with prior similar work, such as the GPPVAE? In principle, it should be easy to improve on this baseline."**
> We thank the reviewer for encouraging us to consider more rigorous experimental evaluation. Following this suggestion, we carried out experimental evaluation of the GPPVAE on both the EEG and Jura experiments, the results of which are now included in Appendix F. We find that the GPPVAE achieves worse predictive performance than the GP-VAE in both experiments, which as the reviewer implored, owes to the factor that a mean-field posterior is a poor approximation to the true posterior.
>
> **"How good is the model itself? Here we see results based on amortized inference, but what if we weren't using it? Even though this case would be slow and maybe not able to handle the entire dataset, this can be done in a subset. In general, it is unclear how much is being lost by a) amortized inference; b) sparsity; c) use of deep sets (in this latter case, it could that something is won instead)."**
> We provided results for the EEG and Jura datasets when amortised is not used in Table 1—these appear as 'FF' (free-form). The results show that the use of amortisation does not degrade the predictive performance of the model. Similarly, we compared the performance of the sparse GP-VAE to that of the full GP-VAE (i.e. $M$=256) in Figure 2, which shows that the use of sparsity worsens predictive performance (as expected), yet results in substantial reduction in training time. As discussed above, the use of deep sets (i.e. partial inference networks) improved the performance relative to the use of zero imputation and the standard encoder, suggesting that it too does not degrade performance.
>
> **"In the Japanese Weather experiment, when the standard VAE is used (with no mean imputation), how are the unobserved values being dealt with?"**
> The standard VAE uses FactorNet to handle unobserved values. We apologise for not making this clearer, and have corrected the paper to do so.
>
> **"In the Japanese Weather experiment, the baselines don't seem very strong. A standard VAE and independent GPs don't seem to be particularly suited to this spatiotemporal weather prediction."**
> The baselines are indeed poor relative to the SGP-VAE; however, this reflects the difficulty of this task. Due to the large number of tasks and datapoints per task, the Japanese weather experiment demands the use of amortisation and sparse GP approximations. To the best of our knowledge, no existing multi-output GP model satisfies these requirements and so a comparison cannot be made. Being able to model datasets such as this was a core motivation throughout this project.

---

> > ### Author Response · Authors · 2020-11-18
> > **Response to review #1 (part 2/2)**
> >
> > **"In general, I'm not sure where the advantage seen in these experiments is coming from. For instance, when compared with published results on the same data (e.g., GPAR on EEG), the SGP-VAE doesn't seem to offer a big advantage. In the other experiments, no strong baselines for that specific data seem to be provided."**
> > GPAR represents the current state-of-the-art for multi-output GP models on both the EEG and Jura experiments by a significant margin. The reviewer is correct in stating that the SGP-VAE doesn't offer a big advantage on these small datasets—our intention was to demonstrate the utility of the SGP-VAE on small datasets, and ensure that amortisation, especially in the presence of partially observed data, is not at the expense of predictive performance. The fact that we achieved comparable to state-of-the-art performance confirms this.
> >
> > Whilst GPAR performs well in the small datasets above, it has two key limitations. First, it can only be used with specific patterns of missing data and cannot be used when the missingness pattern is arbitrary. Second, it is not scalable and would require further development to handle the large datasets considered in this paper. In contrast, the SGP-VAE is far more flexible: it handles arbitrary patterns of missingness, and scales to large numbers of datapoints and tasks. Another advantage of the SGP-VAE is that is models $P$ outputs with just $K$ latent GPs. This differs to GPAR, which uses $P$ GPs. Whilst this isn't an issue for the small-scale experiments, it quickly becomes computationally burdensome when $P$ becomes large (e.g. the bouncing ball experiment would demand GPAR to use 100 GPs, whereas the SGP-VAE uses just 2). The true efficacy of the SGP-VAE is showcased in the final two experiments, where the number of datapoints and tasks is large, and the patterns of missingness are random.
> >
> > **"I would be curious to know in which of these datasets the model itself is superior (if we were to skip the encoder and use, say, MCMC for inference) and in which ones it isn't."**
> > For the datasets we consider, performing MCMC for inference in this model would be enormously challenging. Further, it is not immediately clear that using MCMC would result in better predictive performance than the use of the variational approximation. Intuitively, we can expect the model to be superior for settings in which the distribution of observations is non-Gaussian. For example, in the weather experiments the distribution of precipitation exhibits a peak at zero and a single heavy tail extending over larger values. A linear transformation of the latent GPs would be insufficient in modelling this, hence the flexibility of the non-linear NN transformation offers a distinct advantage.

---

### Author Response · Authors · 2020-11-18
**Response to all reviewers**

We are delighted that all three reviewers vote for acceptance, and thank each reviewer for their useful comments. We greatly appreciate the time and effort you have put into reading our paper. We have uploaded a revised version of the paper, incorporating your comments and suggestions.

**Outlining our central contribution**

Despite having consistently positive reviews, we feel that the central contribution of our paper has not been as well appreciated as we had intended. The use of amortised inference in deep generative models (DGM) and sparse approximations in Gaussian process (GP) models have enabled inference in these respective models to scale to the large quantities of data typical of modern datasets. It is therefore critical to deploy both of these innovations is GP-DGMs (the probabilistic model we employ). However, the combination is not straightforward. This arises from the fact that the standard form for the sparse GP approximate posterior, $q(f) = p(f|u)q(u)$ with $q(u) = \mathcal{N}(u; m, S)$ where $m$ and $S$ are variational parameters, does not decompose into a product over $N$ factors and is therefore not amendable to per-datapoint amortisation — that is, $m$ and $S$ must be optimised as free-form variational parameters.

There are several 'naïve' approaches to achieving per-datapoint amortisation that were not as principled, or successful, as our final approach. For example, we could decompose $q(u)$ into the prior $p(u)$ multiplied by the product of $M$ approximate likelihoods over each inducing point, with the approximate likelihoods equal to the product of per-datapoint approximate likelihoods. The per-datapoint approximate likelihoods depend on both the observation $y_n$ and the distance from the inducing point, $x_n - z_m$ — an inference network can be used to map from $(y_n, x_n - z_m)$ to the parameters of the corresponding approximate likelihood. Whilst this approach works, it is somewhat 'ad hoc' and required passing each datapoint/inducing point pair through an inference network which scales very poorly.

We believe the most exciting and significant contribution of our paper is to show that the optimum approximate posterior over inducing points, $q(u)$, decomposes into the prior $p(u)$ multiplied by product of $N$ factors, one for each datapoint (see Appendix 1). This naturally lends itself to per-datapoint amortisation by using an inference network to map from each observation to the parameters of the approximate likelihood. It is worth emphasising the generality of this result: optimality holds for any GP model with arbitrary point-wise likelihood and independent GP priors. This includes GP classification models and the GP regression network.

Having developed the framework for performing scalable inference in the model, we then sought to address the remaining difficulties encountered when modelling spatio-temporal datasets. In particular, the common occurrence of missing data rendered 'standard inference networks' not fit for use. An elegant solution, proposed by Ma et al. (2018) in the partial VAE, is to reinterpret partial observations as a set and use permutation invariant set functions as inference networks. Partial inference networks achieve just this, and provide an alternative to the commonly used `ad-hoc' solution of setting missing values to zero.

**Summary of changes in the revised paper**

Here we summarise the main changes we have made to the paper in response to the reviews:
* We have included a new section (section 2.1) outlining the need for both sparse approximations and amortised inference in the datasets we consider, and discuss the challenges associated with achieving this.
* We elaborate on the three major shortcomings of existing sparse approximations and how our approach overcomes these at the end of section 2.2.
* We have included results for the GPPVAE on the Jura and EEG experiments in appendix F, which demonstrate superior predictive performance of the GP-VAE.
* Finally, we discuss the limitations of GPAR---the current state-of-the-art multi-output GP model on small data sets which do not have missing data---and how these are addressed by the SGP-VAE at the end of section 4.2.

[Ma et al., 2018] EDDI: Efficient dynamic discovery of high-value information with partial VAE.

---

### Decision · Program_Chairs · 2021-01-07
**Final Decision**

**Decision:**

Reject

**Comment:**

The paper proposes a method for inference in models with GP priors and neural network likelihoods for multi-output modelling, dealing with the problem of scalability and missing data. The paper builds upon previous work on inducing variables for scalability on GP models and inference networks for amortization (reducing the number of parameters to estimate) and dealing with missing data.

There are several concerns about the paper in terms of generality/flexibility of the approach, as the proposed model shares the NN parameters across tasks and the results on the small datasets do not show improvements wrt baseline such as GPAR. The authors’ comments provide somewhat satisfactory replies to these issues. Nonetheless, the major drawback of this paper is its novelty as the ideas on the paper have been explored extensively in the GP literature. Although the authors do make a case for scalability when using inference networks, there are other previous works that perhaps the authors are unaware of, for example, https://arxiv.org/abs/1905.10969   and even more sophisticated inference algorithms than can serve as truly state-of-the-art competing approaches (for example based on stochastic gradient Hamiltonian Monte Carlo, https://arxiv.org/abs/1806.05490).

---

> ### Author Response · Authors · 2021-02-15
> **Response to Area Chair**
>
> We thank the Area Chair for taking the time to review the paper and provide feedback.
>
> We would like to respond to a few key points raised as we believe they could cause an inaccurate characterization of the paper to be perpetuated.
>
> The main contribution of this work is to enable Gaussian Process models with flexible likelihoods to scale to extremely large datasets with large numbers of tasks using a novel form of amortization / inference network that naturally interfaces with pseudo-point approximation methods.
>
> The key contribution is to have an inference / amortization network, that i) is shared across all tasks—just as a standard VAE shares an inference network across all data points—so that the memory overhead does not grow with the number of tasks and the space of parameters that is optimized over is fixed, and ii) naturally interfaces with pseudo-point approximation methods so that each task itself can be large.
>
> In this light, the fact that the amortization network shares neural network parameters across tasks is an essential feature.
> Also note that the generative model we employ does in fact encompass those that have task-specific parameters (therefore we disagree that the approach is limited to sharing parameters across tasks). To see this, note that
> although the neural network which parameterises the likelihood is shared across tasks, it can be modulated in a task-specific manner. Consider the case in which the length-scale of one of the latent GPs is very large—this results in one dimension of the latent variables being fixed across the input domain for any specific task, but it may vary between tasks, and it therefore is equivalent to a task specific parameter that modulates the likelihood function. Rather than obtaining a point-wise estimate of this task-specific parameter, our method enables approximate Bayesian inference.
>
> Several reviewers were concerned about the strength of the GPAR baseline. This was included precisely because it is state-of-the-art on a number of tasks.  However, GPAR is applicable in only a limited number of settings requiring small datasets and restrictions on the structure of missing data. In contrast, we are focused on large datasets, with extensive missing data. So the experiments that use GPAR merely highlight that in cases where GPAR is appropriate, the new method is comparable with the state-of-the-art.
>
> In terms of the specific papers the AC cites:
>
> The first paper (https://arxiv.org/abs/1905.10969) considers a different problem to that considered in our paper, namely that of designing flexible non-Gaussian approximate posterior distributions over functions for a single task. To do this, the authors parameterize the approximate posterior $q(f)$ using a Bayesian neural network (or other distribution over functions) which they refer to as an inference network. However, this is very different usage of the term inference network to ours—we use inference network specifically to refer to an amortization network. The parameters of $q(f)$ in their paper are treated in a ` 'free-form' manner and must be learned separately for each task. So, in this prior work the inference network does not provide cross-task amortisation, unlike our work, and it does not leverage pseudo-point approximations.
>
> The second paper (https://arxiv.org/abs/1806.05490) uses stochastic HMC for inference. HMC and MCMC methods generally scale poorly with the dimensionality of the problem, whereas our inference scheme scales quadratically with the number of inducing points. So for the large-data applications we consider, their approach would be extremely time consuming. Furthermore, the results in the cited paper are not state-of-the-art. For example, the results are generally significantly poorer than those presented in http://proceedings.mlr.press/v48/bui16.pdf and it is interesting to note that Bui et al. use the same functional form for the approximate posterior as our method in their EP approach.  A drawback of the EP approach in the class of generative models we consider, is that it requires numerical integration or sampling of the likelihood, which in the case of neural network likelihoods, would be both extremely expensive and difficult to take gradients through (and therefore difficult to learn)—our approach avoids both of these problems.
>
> Finally, we would like to thank the Area Chair and reviewers for their feedback.  The discussion here has highlighted a number of areas in which we need to improve the presentation of our work, which we will consider in more detail and  address in a future version of the paper.